

# Multispecies expression of coccolithophore vital effects with changing $CO_2$ concentrations and pH in the laboratory with insights for reconstructing $CO_2$ levels in geological history

Goulwen Le Guevel[1][2], Fabrice Minoletti[1], Carla Geisen[2], Gwendoline Duong[3], Virginia Rojas[1] and
Michaël Hermoso[2]

[1]Institut des Sciences de la Terre de Paris (UMR 7193 ISTeP), CNRS, Sorbonne Université, 75005 Paris, France
[2]Laboratoire d'Océanologie et de Géosciences (UMR 8187 LOG), Université du Littoral Côte d'Opale, CNRS, Université de
Lille, 62930 Wimereux, France
[3]Laboratoire d'Océanologie et de Géosciences (UMR 8187 LOG), Université de Lille, CNRS, Université du Littoral Côte
d'Opale, 62930 Wimereux, France

Correspondence to: Michael Hermoso (michael.hermoso@univ-littoral.fr)

**Abstract.** The coccolith sedimentary and micropalaeontological archive has fostered great interest for palaeoclimate applications. Indeed, the geochemistry of coccolith calcite has the potential to reconstruct both palaeo-$CO_2$ concentrations and palaeo-temperature of seawater. Studying coccolith geochemistry aims at better understanding the changes in the vital effect of coccoliths with changes in environmental parameters, especially the carbonate chemistry of seawater. To this aim, we need to deconvolve the biological imprint from the environmental signals recorded in the composition of coccolith biominerals. We have undertaken large-scale culture experiments of four strains of coccolithophores of various sizes and growth rates, grown under eight $CO_2$/pH conditions typifying the long-term $CO_2$ evolution of the Cenozoic Era. We propose an assessment of the expression of the vital effects for *Emiliania huxleyi*, *Gephyrocapsa oceanica*, *Helicosphaera carteri* and *Coccolithus braarudii* with simultaneous changes in Dissolved Inorganic Carbon (DIC) and pH in the medium resulting in variations in $CO_{2\,aq}$ availability to the cells. We have identified a distinct isotopic response of *C. braarudii* to $pCO_2$ levels on either side of the 600 ppmv (pH 7.89) condition. We propose that this discrepancy is the result of a modification of the proton efflux across the plasma membrane (voltage-dependent proton channels). We further show that as the $CO_2$ level rises and pH decreases (from 200 to 500 ppmv and from 8.29 to 7.96 pH units, respectively), a significant increase in $\delta^{13}C_{coccolith}$ of *C. braarudii* is expressed, along with a coeval decrease in $\delta^{13}C_{org}$. The constant physiological parameters of *C. braarudii* (growth rate, PIC, POC) across the 200 to 500 ppmv interval support the idea that the change in $\delta^{13}C_{coccolith}$ is only a consequence of a lower fractionation between dissolved $CO_2$ and organic matter. Meanwhile, the small (less carbon-limited) cells of *E. huxleyi* and *G. oceanica* do not exhibit any change in their carbon vital effects with changes in carbonate chemistry of the environment across the whole $CO_2$ spectrum. Using this new biogeochemical framework, we have established a calibration between $CO_{2\,aq}$ concentration and the differential vital effect ($\Delta\delta^{13}C$) between isotopically-invariant small *G. oceanica* and large coccoliths *C. braarudii*, whose vital effect is $CO_2$-dependent at low $CO_2$. The $CO_2$-$\Delta\delta^{13}C$



transfer equation allows palaeo-$p$CO$_2$ reconstructions based on isotope changes explained by physiological processes, especially at low and medium CO$_2$ levels.

## 1 Introduction

The atmospheric concentration of carbon dioxide ($p$CO$_2$) is a key parameter controlling global climate through its radiative forcing on Earth's surface temperatures. The reconstruction of past $p$CO$_2$ is challenging to handle as many caveats exist in the palaeo-CO$_2$ barometry methods. The magnitude of the carbon isotope fractionation between membrane lipids called alkenones produced by the coccolithophores – found in sediments – and ambient aqueous CO$_2$ constitutes the basis of the foremost palaeo-$p$CO$_2$ proxy, referred to as $\varepsilon_{\text{p-alk}}$ (Pagani et al., 2010; Zhang et al., 2019, 2020). This approach has led to significant advances in the understanding of the past CO$_2$ concentration, but its application still has some limitations. Conversely to the coccolith archive, alkenone molecules are not ubiquitously preserved in marine sediments and the assessment of $p$CO$_2$ involves a cascade of calculations requiring assumptions made on the $\delta^{13}$C value of CO$_{2\,\text{aq}}$. Furthermore, this method has recently been shown to overestimate $p$CO$_2$ below 270 ppmv (Badger et al., 2019), potentially leading to underestimated $p$CO$_2$ above 270 ppmv. Alongside this proxy, the boron isotope fractionation in foraminifera tests has increasingly been used to reconstruct palaeo-pH (Rae et al., 2021). Combined with another parameter of the carbonate chemistry system as DIC or alkalinity, and with temperature, palaeo-pH can be used for palaeo-$p$CO$_2$ reconstructions (Foster, 2008; Sanyal et al., 1995). This proxy also suffers from uncertainties due to the evaluation of $\delta^{11}$B of past seawater among others unknown factors (Klochko et al., 2006; Tripati et al., 2011).

In this study, we investigated the fractionation of carbon and oxygen isotopes in phytoplanktonic organic matter and coccolith calcite driven by changes in the carbonate chemistry of coccolithophores culture media. We have conducted *in vivo* culture experiments, that provide a means to constrain the cellular and isotopic responses of coccolithophores to environmental changes. Our culture experiments were conducted to quantify the isotopic departure of coccoliths from inorganic calcite (vital effect). The latter only depends on physico-chemical parameters of the environment such as $\delta^{18}$O$_{sw}$, $\delta^{13}$C$_{\text{CO2}}$, temperature, salinity, and pH (Zeebe and Wolf-Gladrow, 2001). However, when calcite is biomineralised intracellularly, biological parameters such as growth rate, cell size, and the PIC/POC ratio - which refers to the distribution of carbon between particulate organic carbon (POC) and particulate inorganic carbon (PIC) produced by calcifying organisms - also influence this fractionation (Dudley et al., 1986; McClelland et al., 2017; Rickaby et al., 2010). The causes behind the changes in the isotopic fractionation between the different cellular compartments are to be investigated at various stages of carbon fixation. Aqueous CO$_2$, the main form of DIC entering coccolithophorid cells, diffuses through the plasma membrane via passive diffusion (Gutknecht et al., 1977). CO$_2$ diffusion is governed primarily by Fick's first law, but also by the specific conditions of the cell boundary layer on the external side of the plasma membrane (Reinfelder, 2011; Wolf-Gladrow and Riebesell, 1997). In addition to passive CO$_2$ diffusion, certain taxa of coccolithophores such as *Emiliania* possess CCMs (Carbon Concentration Mechanisms), allowing the intake of HCO$_3^-$ through active transmembrane transport



(Bach et al., 2013; Parker and Boron, 2013; Romero et al., 2004). As $HCO_3^-$ is enriched in $^{13}C$ compared to $CO_2$, the greater acquisition of $HCO_3^-$, the higher $\delta^{13}C$ values in the intracellular DIC pool and ultimately the higher $\delta^{13}C$ of coccolith calcite. Once inside the cell, carbon is utilised by two main pathways: the fixation into organic matter via photosynthesis in chloroplasts and precipitation of calcite forming coccoliths within the coccolith vesicle. Carbon fixation into organic matter by the enzyme RuBisCO (ribulose 1,5-bisphosphate carboxylase/oxygenase) (Ellis, 1979) leads to fractionation as it

preferentially fixes the lighter isotopes ($^{12}C$) (Laws et al., 2001; Popp et al., 1998). Consequently, organic matter is relatively depleted in $^{13}C$ (very negative $\delta^{13}C_{org}$) compared to the carbon source and compared to the intracellular DIC pool (Bidigare et al., 1997; Guy et al., 1993; Rau et al., 1996). The fractionation induced by RuBisCO is one component of a more general fractionation between the $CO_{2\,aq}$ from the external environment and the organic matter synthesised within the cell, which leads to an enrichment, noted $\varepsilon_p$ (Jasper and Hayes, 1990; Pagani et al., 1999). In addition to being dependent on the

fractionation associated with RuBisCO, $\varepsilon_p$ is influenced by physiological parameters such as growth rate, cell size, and the presence of CCMs (Laws et al., 1995; Popp et al., 1998).

In this work, we provide both $\delta^{13}C_{coccolith}$ and $\delta^{13}C_{org}$ values with changes in $p$CO$_2$ and pH to explain the mechanisms responsible for the carbon isotope fractionation within the cell. The present study stems from several lines of evidence that the isotopic offset between coccoliths of different sizes conveys a specific $p$CO$_2$ signal as shown in sedimentary records by

the work of Bolton and Stoll (2013). This empirical observation has been repeatedly reported in culture data (Hermoso, 2015; Hermoso et al., 2016b; McClelland et al., 2017; Rickaby et al., 2010). This approach used to derive palaeo-CO$_2$ concentrations still needs to be fully constrained, although a proof of concept exists (Bolton and Stoll, 2013; Godbillot et al., 2022; Hermoso et al., 2016b, 2020; Tremblin et al., 2016). Modelling studies fed by culture data have identified and quantified the main forcing parameters behind the magnitude of carbon isotope vital effect in coccolith calcite: growth rate,

cell size, the partitioning of CO$_2$ in particulate inorganic matter and particulate organic matter (PIC/POC ratio), among other ancillary parameters (McClelland et al., 2017).

Four strains of geological-relevant coccolithophores, representing a wide diversity of growth rate and coccolith and cell sizes, were cultured to study how the composition of the culture medium (DIC, pH) influences the magnitude of coccoliths vital effects and to establish transfer functions between these vital effects and the aqueous CO$_2$ concentrations and pH. In

contrast with previous culture studies and to accurately mimicking the carbonate chemistry of the ocean through the Cenozoic Era, we cultured these calcifying microalgae with coupled $p$CO$_2$ and pH perturbations of the medium. The culture conditions varied from 200 ppmv/8.29 pH units (pre-industrial) to 1400 ppmv/7.55 pH units (thought to represent the Mid-Eocene levels) keeping all other parameters constant. Particular attention has been paid to conditions similar to those of the last 12 Myrs (narrow step of 100 ppmv for $p$CO$_2$ between 200 and 500 ppmv) because of the scarcity of available data.



## 2 Materials and methods

### 2.1 Strains

We cultured four coccolithophore strains kindly provided by the biological station of Roscoff: RCC1200 *Coccolithus braarudii*, RCC1323 *Helicosphaera carteri*, RCC1314 *Gephyrocapsa oceanica*, and RCC1256 *Emiliania huxleyi* (Figure 1). These species represent a wide range of coccolith taxa with various coccosphere and cell sizes. The rationale of this choice is that they belong to the most abundant groups found in Neogene pelagic sediments (Bolton et al., 2012; Claxton et al., 2022). If comparable with our own study (mode of culture, temperature, pH etc…), the results of previously-published biogeochemical work on the same species will be presented along with our own data in the figures.

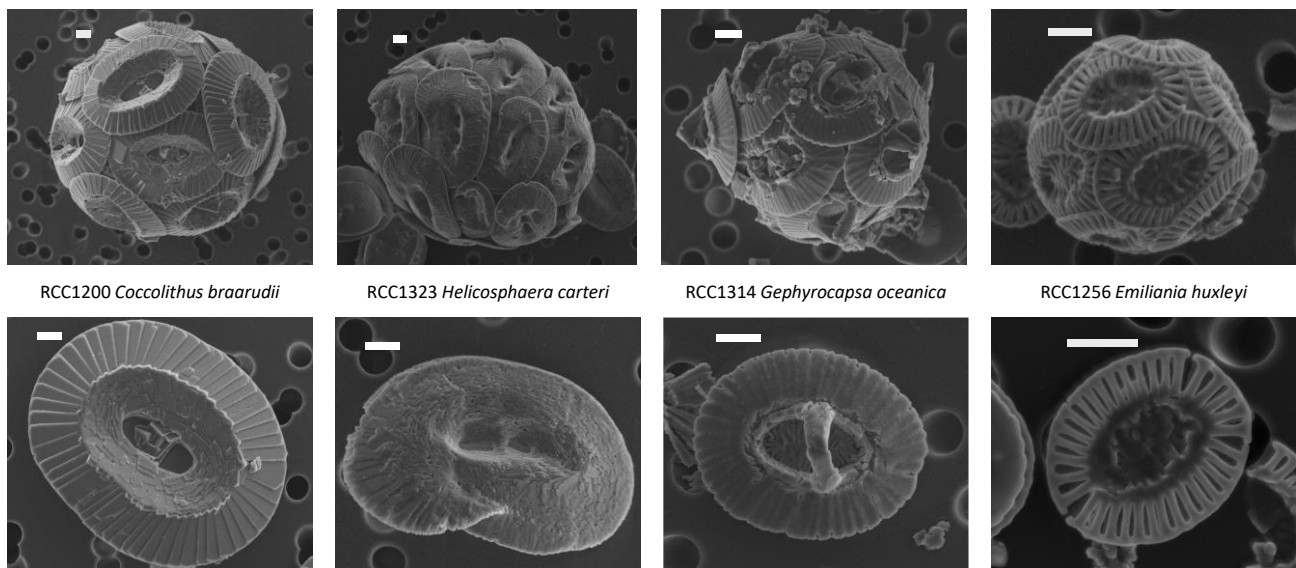

RCC1200 *Coccolithus braarudii*   RCC1323 *Helicosphaera carteri*   RCC1314 *Gephyrocapsa oceanica*   RCC1256 *Emiliania huxleyi*

**Figure 1: Scanning electron microscope (SEM) images of the four studied strains. Top images show the coccoliths and bottom images show the coccospheres. The scale bars represent 1 µm.**

The cultures were undertaken at the Maison de la Recherche en Environment Naturel (ULCO - LOG) in Wimereux (Northern France) in 2023 – 2024.

### 2.2 Medium preparation

Our culture experiments were designed to represent pH and $p$CO$_2$ from the last greenhouse period (Eocene) to the preindustrial Holocene oceanic conditions (Rae et al., 2021; Sosdian et al., 2018). Variable $p$CO$_2$ between 200 and 1400 ppmv at constant alkalinity of 2300 µmol/kg have been used to calculate the target pH (total scale) with CO2sys program (Bakker et al., 2016) See table 1 for details on the chemical parameters of the culture medium.





Artificial seawater with a salinity of 34 psu was prepared following the ESAW recipe of Berges, Franklin and Harrison, (2001). The use of artificial seawater was preferred over natural seawater in order to facilitate the production of medium with $CO_2$ level under 400 ppmv. Phosphate, nitrate, silica, Fe-EDTA, trace metals and vitamins were added to produce K/2 media according to Keller et al. (1987). The DIC concentration was reached adding different amounts of $NaHCO_3$ (Sigma – batch CAS 144-55-8). The pH was adjusted with HCl and NaOH addition until the target pH was reached. This treatment led to a change in the relative abundance of the DIC species (see Bjerrum plot of carbonate speciation versus pH, Zeebe and Wolf-Gladrow, 2001). Lastly, the medium was sterilised with a 0.22 µm filtration step and stored without headspace in amber-coloured bottles at 15 °C in the dark.

## 2.3 Culture growth

The coccolithophore strains were first acclimatised to the medium during about 10 generations in 25 cm² polystyrene flasks. The cells were then inoculated in triplicate experiments in culture bottles of increasing size (75 cm² polystyrene flasks, 600 mL and 2300 mL polycarbonate bottles) until the target cell number of the culture was reached. We grew the cultures until reaching a sufficient biomass for our analyses (between 2 and 10 mg for the small cells *E. huxleyi* and *G. oceanica* and between 50 and 60 mg for the large cells *H. carteri* and *C. braarudii*), while ensuring that the cultures remained adequately diluted.

The cell cultures were maintained at constant temperature (15 °C) with a 14/10 day-length cyclicity and an irradiance of 150 µmol.photons.m$^{-2}$.s$^{-1}$. The cells were regularly resuspended to avoid cell clustering and kept at relatively low concentrations to maintain homogeneous carbon bioavailability and light access during all the experiment. A daily control of the culture health was made with a reverse optical microscope under x400 magnification. A final control of the coccoliths was made after the harvest with a Zeiss Supra 55 VP SEM at Sorbonne University.

Cell numeration and coccosphere sizes were measured using a Beckman Coulter Counter Multisizer 4e calibrated with 10.16 µm latex beads. The measurements were always performed at the same time of the day (8:30 to 10:00 am) to avoid biases related to the growth of the cell during the day phase.

The growth rate µ in day$^{-1}$ of a microorganism culture corresponds to the increase of the cell number by time units commonly calculated with the formula:

$$\mu \ (day^{-1}) = \frac{ln(c_{final}) - \ln(c_{initial})}{t_{final} - t_{initial}}, \tag{1}$$

where c is the cell concentration and $t_{final}$-$t_{initial}$ is the number of days between the initial and the final cell concentration measurements. To take into account all the concentration measurements, the µ was derived from the slope of the linear regression of the function $f(t_x$-$t_{initial}) = ln(c_x)$ where x is the day of the measurement of $c_x$.



**2.4 Isotopic reference of the culture medium**

The $\delta^{13}C$ of the Dissolved Inorganic Carbon ($\delta^{13}C_{DIC}$) and the DIC concentrations were measured in 11 samples of various $CO_2$/pH with an Isotope and Gas Concentration Analyzer Picarro G2131-i coupled with an Apollo SciTech DIC-$\delta^{13}C$ analyser AS-D1 at LOCEAN laboratory (Sorbonne University). The medium samples were preserved in 500 mL glass bottle with ground neck with 0,3 mL saturated $HgCl_2$ solution. The measurements are calibrated with in-house standards. Since we used the same $NaHCO_3$ powder batch for the different experiments, all the culture media had the same $\delta^{13}C_{DIC}$ values (-

12.17‰ V-PDB ($\pm 0.07$)).

The $\delta^{18}O$ of seawater was measured by cavity ring down spectroscopy (CRDS) using a Picarro instrument (model L2130-i Isotopic $H_2O$) at LOCEAN, Sorbonne University. The in-house standards (freshwater) are calibrated using IAEA references V-SMOW (Vienna Standard Mean Ocean Water) and GISP (Greenland Ice Sheet Precipitation). The $\delta^{18}O$ of the inorganic reference ($\delta^{18}O_i$) in ‰ V-PDB is calculated from the $\delta^{18}O_{sw}$ and temperature (15 °C) according to the equation of Kim and

O'Neil 1997 modified by Tremblin, et al. (2016). The $\delta^{18}O_{sw}$ slightly changed between successive culture campaigns: -6.44‰ V-SMOW ($\pm 0.03$) for *C. braarudii* and *G. oceanica*, -6.67‰ V-SMOW ($\pm 0.03$) for *H. carteri* and -6.36‰ V-SMOW ($\pm 0.15$) for *E. huxleyi*).

The oxygen vital effects are calculated as the difference between the $\delta^{18}O$ of the coccoliths ($VE^{18}O$) in ‰ V-PDB and the $\delta^{18}O$ of a theoretical inorganic calcite ($\delta^{18}O_i$) in ‰ V-PDB:

$$VE^{18}O = \delta^{18}O_{coccoliths} - \delta^{18}O_i ,\qquad\qquad\qquad\qquad\qquad (2)$$

**2.5 Isotopic analysis on calcite**

Two methods were used to collect the culture residues for further isotopic measurements of coccolith calcite:

- RCC1200 *C. braarudii* and RCC1323 *H. carteri* culture residues were centrifuged during 15 minutes at 4500 rpm in 500 mL centrifuge bottles. After removing the supernatant, samples were rinsed with three cycles of centrifugation with

replacement of the supernatant with neutralised demineralised water and homogenisation between each cycle. The rinsing is employed to remove the salt from the culture water. The culture residues were finally dried at 40 °C after the removal of the last supernatant.

- RCC1256 *E. huxleyi* and RCC1314 *G. oceanica* samples were collected and rinsed three times with neutralised demineralised water onto polycarbonate filtering membranes, and then dried in a desiccator. We applied a different

harvesting method for those strains because of their small size, that induced too much material loss during the centrifugation process. All culture residues were stored at 5 °C after harvest.

Isotopic measurements on calcite were performed at the ISTeP laboratory (Sorbonne University). Between 30 and 60 µg of the samples were digested with 100% phosphoric acid at 70°C in pre-evacuated vials using a Kiel IV carbonate device. The evolved $CO_2$ was purified in a cryogenic trapping system and carbon and oxygen isotope compositions were measured in an

isotope-ratio mass spectrometer DELTA V advantage (Thermo Scientific) with a dual inlet introduction system. The carbon



and oxygen isotope composition were expressed in the delta notation as a value relative to the Vienna Pee Dee Belemnite (V-PDB) and reported in permil (‰). $\delta^{13}C$ and $\delta^{18}O$ values were calibrated using NBS-19 and NBS-18 international standards. Internal reproducibility and accuracy were monitored by replicate analysis of in-house calcite standard Marceau ($\delta^{13}C$ = 2.12‰ and $\delta^{18}O$ = -1.87‰), being the measured values for $\delta^{13}C$ and $\delta^{18}O$ 2.10 ±0.08‰ (1$\sigma$) and -1.89 ±0.10‰ (1$\sigma$),

respectively. The external reproducibility, obtained from replicate analysis of the samples, is better than 0.05‰ (1$\sigma$) for $\delta^{13}C_{coccolith}$ and 0.10‰ (1$\sigma$) for $\delta^{18}O_{coccolith}$. Transfer equations between coccolith isotope ratios and $CO_2$ levels are proposed in this study. The residual errors are evaluated through Monte Carlo analysis code with 1,000,000 iterations and an uncertainty of 0.17‰ for the differential vital effect between small and large coccolithes $\Delta\delta^{13}C_{small-large}$ (1$\sigma$).

## 2.6 Carbon isotope analysis of the organic matter

The culture residues of *C. braarudii*, *G. oceanica* and *E. huxleyi* were gathered onto glass microfiber filters (GFF), rinsed three times with neutralised demineralised water and then stored at -18 °C. We then gently scratched the superficial part of the GFF to collect all the culture residue. The culture residues were decarbonated with hydrochloric acid 2N overnight. The samples were then rinsed to eliminate the excess of hydrochloric acid and released $Ca^{2+}$ and alkalinity, and then dried at 35 °C. Between 10 and 50 µg of decarbonated culture residues (only organic matter) were weighted and loaded into tin

capsules.

The carbon isotope composition of the organic matter was measured at the Stable Isotope Geochemistry laboratory at IPGP (Institut de Physique du Globe de Paris). A Flash EA 1112 elemental analyser coupled in continuous helium flow to an isotope ratio mass spectrometer Thermo Fisher Scientific DELTA V Plus was used for the *C. braarudii* samples cultured between 200 and 500 ppmv (8.29 and 7.96 pH units). For *G. oceanica* and *E. huxleyi* for all culture conditions, and for *C.*

*braarudii* between 500 and 1400 ppmv (7.96 and 7.55), we used an Elementar vario PYRO cube analyser coupled with the same spectrometer DELTA V Plus at IPGP. Three organic-rich internal standards, calibrated against international standards, were used to calculate the $\delta^{13}C$ values of the samples, reported in ‰ with respect to V-PDB. Replicate analysis of standards yielded an internal reproducibility better than 0.10‰ (1$\sigma$). Accuracy was assessed by measuring an in-house Tyrosine standard ($\delta^{13}C$ = -23.23‰), for which the estimated error was better than 0.20‰. The external reproducibility for $\delta^{13}C_{org}$,

obtained from replicate analysis of the samples, is better than 0.07%.

## 2.7 Carbon content analysis

The PIC:POC ratios of the four strains studied under various $p$CO$_2$/pH conditions were measured at the Wimereux marine station using a Thermo Fisher Flash 2000 elemental analyzer. The culture residues were collected and stored in the same way for analyses of the $\delta^{13}C_{org}$. The calibration used for reconstructing carbon content was done with various amounts of

acetanilide with known amount of carbon and nitrogen. Samples for POC analysis were previously acidified. The filters were encapsulated in a tin disk before analysis.



| Culture conditions | Temperature (°C) controlled (incubator) | Salinity (psu) measured | Light period length (h) controlled (incubator) | Dark period length (h) controlled (incubator) | pH (total scale) adjusted | TA (µmol/kg) calculated (CO2sys) | DIC (µmol/kg) calculated (CO2sys) | $CO_2$ (µmol/kg) calculated (CO2sys) | $HCO_3^-$ (µmol/kg) calculated (CO2sys) | $CO_3^{2-}$ (µmol/kg) calculated (CO2sys) | $pCO_2$ (ppmv) initial target |
|---|---|---|---|---|---|---|---|---|---|---|---|
| 1 | 15 | 33.86 | 14 | 10 | 8.29 | 2256.0 | 1913.6 | 7.51 | 1669.3 | 236.9 | 200 |
| 2 | 15 | 33.86 | 14 | 10 | 8.15 | 2275.0 | 2011.6 | 11.27 | 1813.9 | 186.5 | 300 |
| 3 | 15 | 33.86 | 14 | 10 | 8.04 | 2248.7 | 2042.2 | 15.03 | 1877.4 | 149.8 | 400 |
| 4 | 15 | 33.86 | 14 | 10 | 7.96 | 2272.4 | 2100.3 | 18.78 | 1951.9 | 129.6 | 500 |
| 5 | 15 | 33.86 | 14 | 10 | 7.89 | 2272.2 | 2128.8 | 22.54 | 1993.6 | 112.6 | 600 |
| 6 | 15 | 33.86 | 14 | 10 | 7.73 | 2269.0 | 2183.6 | 33.81 | 2068.9 | 80.9 | 900 |
| 7 | 15 | 33.86 | 14 | 10 | 7.61 | 2246.3 | 2199.7 | 45.08 | 2092.6 | 62.0 | 1200 |
| 8 | 15 | 33.86 | 14 | 10 | 7.55 | 2262.2 | 2233.8 | 52.59 | 2126.3 | 54.9 | 1400 |

**Table 1: Experimental parameters of the mediums. The salinity has been measured in the initial medium. The temperature and the light and dark period length are controlled by incubator set up. The total alkalinity (TA) and the DIC, $CO_2$, $HCO_3^-$ and $CO_3^{2-}$ concentrations are calculated by CO2sys program. $pCO_2$ and pH correspond to the culture conditions tested.**

## 3 Results

### 3.1 Physiological parameters and carbon content

#### 3.1.1 Growth rates

The growth rates of the four strains do not show any statistical trend with changes in the carbonate chemistry of the medium (*C. braarudii*: r² = 0.04 and p > 0.05, *H. carteri*: r² = 0.08 and p > 0.05, *G. oceanica*: r² = 0.11 and p > 0.05, *E. huxleyi*: r² = 0.38 and p < 0.01) (Figure 2). More than 85% of the data are comprised within a ±15% relative range of the mean growth rate for each strain (coloured band in Fig. 2). *E. huxleyi* and *G. oceanica* have the higher mean growth rate (0.87 d⁻¹ and 0.67 d⁻¹ respectively), while the larger *C. braarudii* and *H. carteri* have a mean growth rate of 0.58 d⁻¹ and 0.24 d⁻¹, respectively (Figure 2). These absolute values are similar to those obtained in previous studies for *E. huxleyi*, *G. oceanica* and *C. braarudii* (Phelps et al., 2021; Rickaby et al., 2010). *C. braarudii* has been reported with decreasing growth rates with rising $CO_2$ levels (Hermoso et al., 2016b; Rickaby et al., 2010).



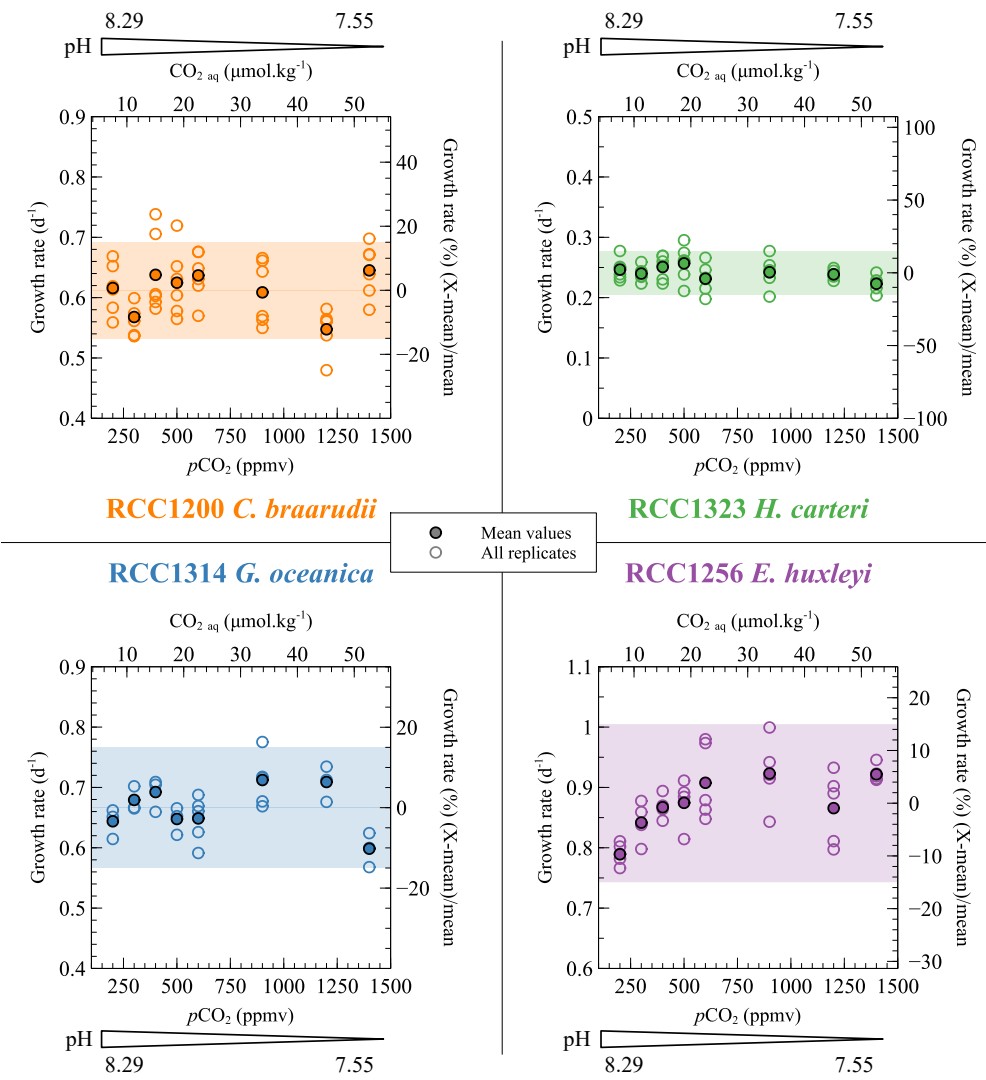

**Figure 2: Specific growth rate (μ) of the four cultured strains with respect to $CO_2$ level and pH of the culture media. The empty dots show all the replicate data and the filled dots denote the mean growth rates. Growth rates are given in $d^{-1}$ (left axis) and as a relative deviation of the mean growth rate of each strain (right axis). The coloured bands correspond to the range of ±15% of variation from the mean growth rate. More than 85% of the data are comprised within this range.**

### 3.1.2 Coccosphere sizes

The sizes of the coccospheres – i.e., the coccolithophore exoskeletons surrounding the cells, composed of coccoliths – approximate the evolution of cell size. The evolution of this parameter among the different strains varies with the $CO_2$/pH treatments. *C. braarudii* displays the largest coccospheres amongst the cultured strains. The mean coccosphere diameter of *C. braarudii* increases from 16.7 to 18.4 μm with increasing $CO_2$ levels from 200 to 600 ppmv and decreasing pH from 8.29



to 7.89 units of pH. The +1.70 µm increase in the coccosphere size represent a relative variation of 10.2% ($r^2 = 0.86$, p <
0.01). For higher $p\text{CO}_2$ and lower pH, the distribution of the coccosphere sizes is not linked with the carbonate chemistry
(Figure 3). The size of *H. carteri* presents a hyperbolic trend ($r^2 = 0.72$, p < 0.01) of the coccosphere size with increasing
$\text{CO}_2$ level and decreasing pH (+0.69 µm from 200 ppmv/8.29 pH units to 600 ppmv/7.89 pH units, and -1.05 µm from 600
ppmv/7.89 pH units to 1400 ppmv/7.55 pH units) (Figure 3). The size of the coccospheres produced by *G. oceanica*
decreases from 8.0 to 7.5 µm with increasing $\text{CO}_2$ levels from 200 to 1200 ppmv and decreasing pH from 8.29 to 7.55 (-0.51

µm, a relative variation of 6.4%, $r^2 = 0.67$, p < 0.01) (Figure 3). The coccosphere size of the small species *E. huxleyi* exhibit
a +0.22 µm increase with increasing $p\text{CO}_2$ and decreasing pH until 600 ppmv/7.89 pH units (a relative variation of 4.4%, $r^2$
= 0.77, p < 0.001). At higher $p\text{CO}_2$ and lower pH conditions, the coccosphere size is constant ($r^2 = 0.04$, p > 0.05) (Figure 3).
These absolute values and trends in coccosphere size with changes in $\text{CO}_2$ level and pH are in line with previously reported
coccosphere and cell sizes from previous culture studies (Hermoso, 2015; Hermoso et al., 2014, 2016b; Phelps et al., 2021;

Rickaby et al., 2010).



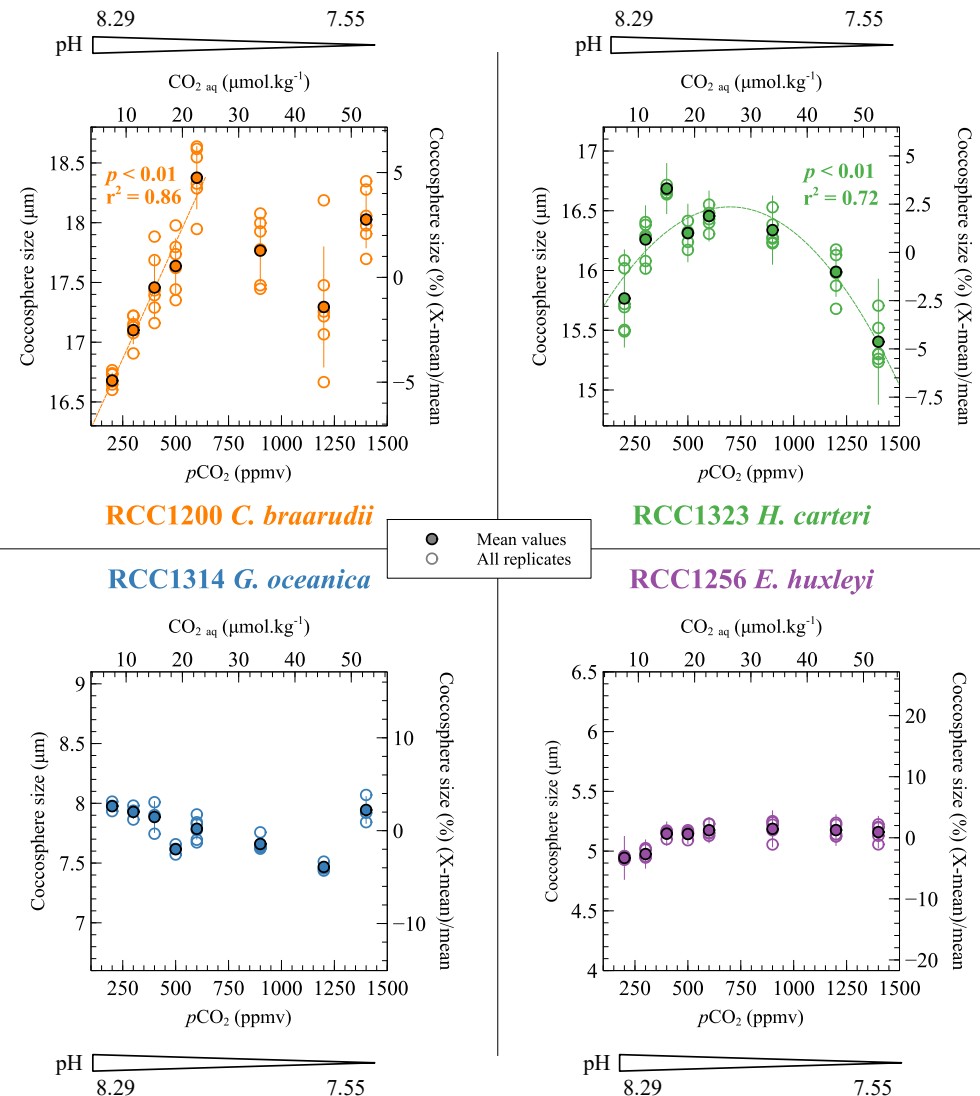

**Figure 3: Evolution of the coccosphere sizes of the four cultured strains as a function of the $CO_2$ and pH culture conditions. The coccosphere sizes are given as the mean diameter of the coccospheres. They are given in µm (left axis) and as a percent relative to the mean coccosphere size of each strain (right axis). The empty points show the replicate data and the filled points show the mean coccosphere sizes.**

### 3.1.3 PIC:POC ratios

The PIC:POC ratio allows quantifying the respective allocation of carbon in its two fixation pathways (photosynthesis and calcification), which has isotopic implications on organic matter and coccoliths. The PIC:POC ratios of the four species do not show any statistically significant trend with changes in carbonate chemistry. Among the species cultured, *C. braarudii*





has the highest PIC:POC ratio (mean value of $2.23 \pm 0.49$), followed by *H. carteri* (mean value of $1.44 \pm 0.35$), *G. oceanica* ($0.98 \pm 0.20$) and *E. huxleyi* ($0.50 \pm 0.09$) (Figure 4). Those data are consistent with previously published PIC:POC data (Krug et al., 2011; Langer et al., 2006; Müller et al., 2010; Rickaby et al., 2010; Riebesell et al., 2000).


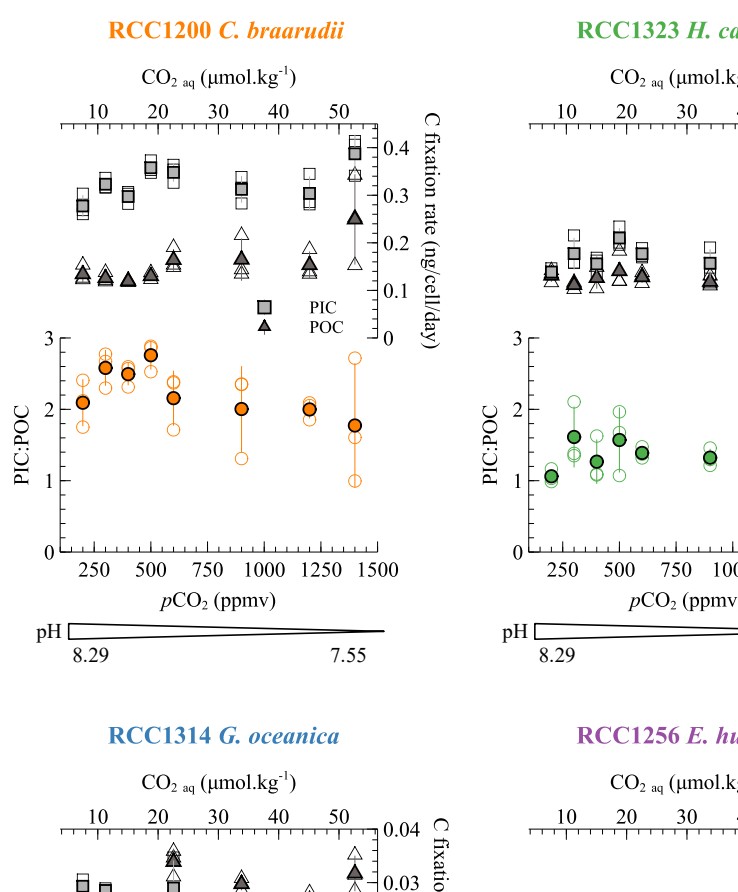

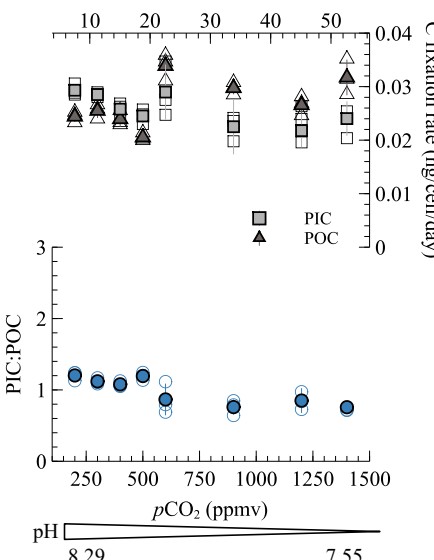

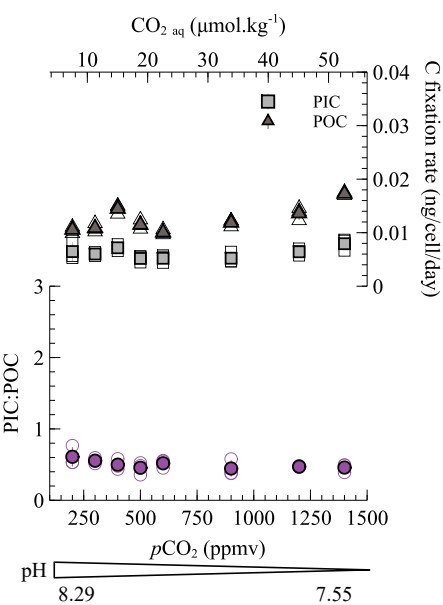



**Figure 4: Particulate inorganic (square) and organic (triangle) carbon in ng per cell for each of the four species through the $pCO_2$ interval of the study (lower x-axis) and the aqueous $CO_2$ (upper x-axis). The PIC:POC ratios are also shown. The empty dots are all the replicate data and the filled points are the mean values for each species/condition.**


### 3.2 Carbon isotope ratios of coccolith calcite

The species *E. huxleyi* does not show any significant trend in $\delta^{13}C_{coccolith}$ with changes in $pCO_2$ and pH ($r^2 = 0.16$, $p > 0.05$). The taxon *H. carteri* exhibits a minor increase of less than 0.5‰ in $\delta^{13}C_{coccolith}$ with increasing $CO_2$ level et decreasing pH ($r^2 = 0.49$, $p < 0.001$) (Figure 5). A key feature of the dataset pertains to the biogeochemical response of *C braarudii*. The

distribution of the isotope data with changes in carbonate chemistry can be divided into two distinct trends for *C. braarudii* and *G. oceanica*. The first trend corresponds to low $CO_2$ level from 200 to 500 ppmv and high pH from 8.29 to 7.96. The second stage corresponds to high $CO_2$ level from 600 to 1400 ppmv and low pH from 7.89 to 7.55 pH units. At low $CO_2$ levels and high pH, the $\delta^{13}C_{coccolith}$ of *C. braarudii* increases with increasing $pCO_2$ and decreasing pH (+2.41‰ V-PDB, a relative variation of 14%, $r^2 = 0.83$, $p < 0.001$). Regarding *G. oceanica*, it exhibits a small $\delta^{13}C_{coccolith}$ increase with

increasing $pCO_2$ and decreasing pH (+0.41‰ V-PDB, a relative variation of 3%, $r^2 = 0.61$, $p < 0.01$). Compared to low $pCO_2$ levels and high pH, the $\delta^{13}C_{coccolith}$ of *C. braarudii* increases with a less steep slope at high $CO_2$ levels and low pH (+0.92‰ V-PDB, a relative variation of 7%, $r^2 = 0.90$, $p < 0.001$). The $\delta^{13}C_{coccolith}$ of *G. oceanica* is steady at high $pCO_2$ and low pH ($r^2 = 0.10$, $p > 0.05$).

These $\delta^{13}C$ data do not exhibit any consistent trend with changes in growth rate, nor with PIC and POC. *G. oceanica* and *E.*

*huxleyi* show a carbon vital effect similar to that obtained in the work of Rickaby et al. (2010) and Hermoso et al. (2016). It is also similar to the dataset published in McClelland et al. (2017). The vital effects of *C. braarudii* in this study have a similar evolution with previously published results (Hermoso et al., 2016b; Rickaby et al., 2010) with a 1 ‰ shift towards more negative values (Figure 5).



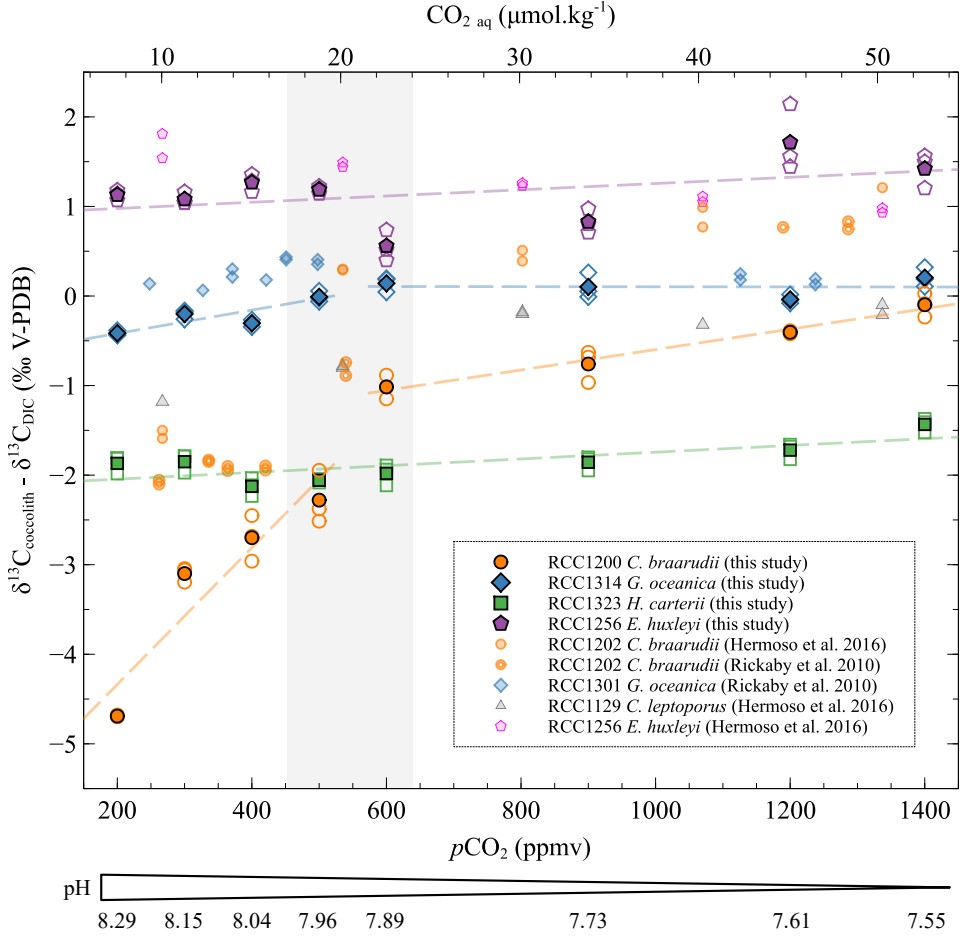


**Figure 5: Evolution of the carbon isotopic ratios of the four studied strains with increasing $CO_2$ concentration and decreasing pH. The data are represented as the $\delta^{13}C$ difference between coccoliths and DIC. The empty points are all the replicate data and the filled points are the means (circles:** *Coccolithus braarudii*, **RCC 1200; diamonds:** *Gephyrocapsa oceanica*, **RCC 1314; squares:** *Helicosphaera carteri*; **pentagons:** *Emiliania huxleyi*). **The grey band represents the carbonate chemistry condition where a shift in**
**the carbon and oxygen isotope ratio occurs (between 500 and 600 ppmv). The pH indicated below only refers to the pH of the cultures of this study. The sources of previously reported data (smaller symbols) are inset.**

### 3.3 Oxygen isotope ratios of coccolith calcite

The $\delta^{18}O_{coccolith}$ of *G. oceanica* is constant regardless ambiant conditions ($r^2 = 0.02$, $p > 0.05$), as for $\delta^{18}O_{coccolith}$ values of *E.*
*huxleyi* ($r^2 = 0.15$, $p > 0.05$). The mean magnitude of the oxygen vital effects of *G. oceanica* and *E. huxleyi* are positive (+1.05‰ and +1.92‰, respectively), in good agreement with previously published data (Dudley et al., 1986; Hermoso et al., 2016b, a; Rickaby et al., 2010; Stevenson et al., 2014; Ziveri et al., 2003).





As already shown in the literature, *H. carteri* have a $\delta^{18}O_{coccolith}$ close to the inorganic reference (mean $\delta^{18}O_{coccolith}$ of +0.42‰) (Ziveri et al., 2003). Indeed, at low $CO_2$ concentrations and high pH, the $\delta^{18}O_{coccolith}$ of *H. carteri* is slowly

decreasing (-0.22‰, a relative variation of 4%, $r^2 = 0.57$, $p < 0.01$). From 600 ppmv and 7.89 pH units, the $\delta^{18}O_{coccolith}$ of *H. carteri* is stable ($r^2 \approx 0$, $p > 0.05$) (Figure 6).

The $\delta^{18}O_{coccolith}$ of *C. braarudii* is constant at low $pCO_2$ and high pH with a mean value of $-0.81 \pm 0.13$‰ V-PDB and is also constant at high $pCO_2$ and low pH with a mean value of $-0.37 \pm 0.12$‰ V-PDB. A significant difference in the *C. braarudii* $VE^{18}O$ is registered between high and low $CO_2$ ambient conditions in the culture medium. This +0.5‰ shift occurs between

500 and 600 ppmv. While we present evidence of a negative vital effect of coccoliths for *C. braarudii*, previous studies have shown a $\delta^{18}O_{coccolith}$ close to that of the inorganic for this species. (Hermoso, 2015; Hermoso et al., 2016a; Rickaby et al., 2010; Stevenson et al., 2014).

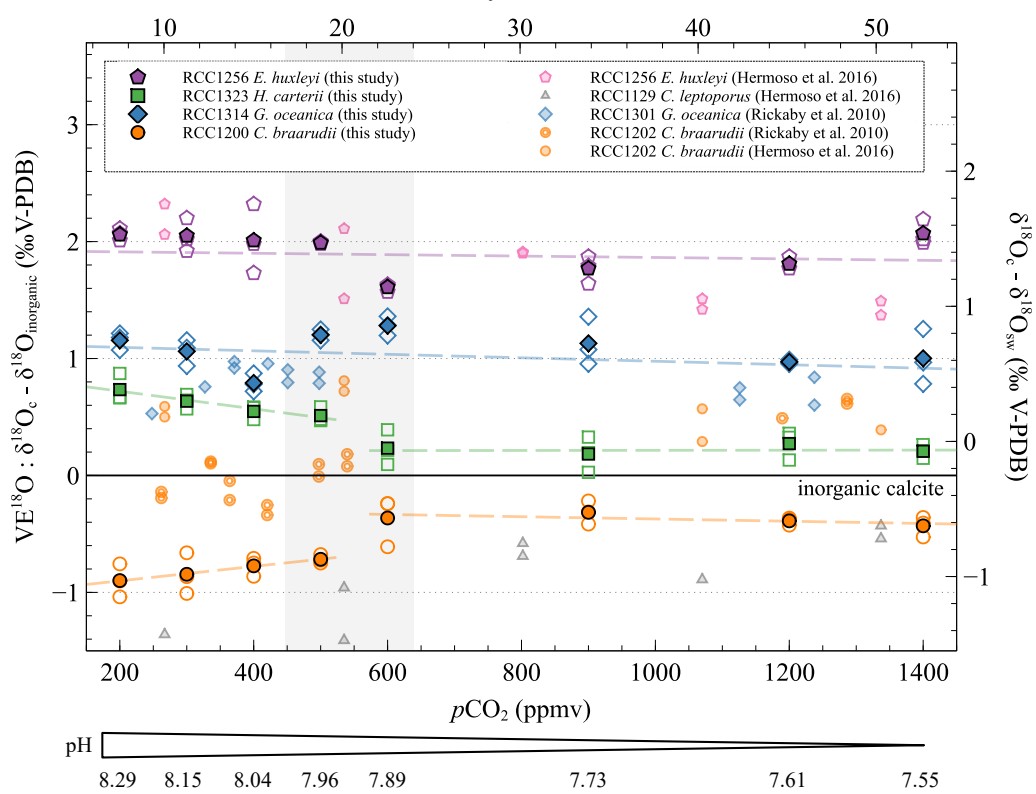

**Figure 6: Evolution of the oxygen isotopic ratios of the four studied strains with increasing $CO_2$ concentration and decreasing pH. The data are represented as the vital effect (left axis) and as the isotopic difference between the seawater and the coccoliths $\delta^{18}O$ (right axis). The empty points are all the replicate data and the filled points are the means (circles:** *Coccolithus braarudii***, RCC 1200; diamonds:** *Gephyrocapsa oceanica***, RCC 1314; squares:** *Helicosphaera carteri***; pentagons:** *Emiliania huxleyi***). The vertical gray band represents the carbonate chemistry condition where a shift in the carbon and oxygen isotope ratio occurs. The black**
**line shows the calculated isotope ratio of an inorganic calcite (see Materials and methods). The pH scales indicated below only refers to the data published in this study. The sources of previously reported data (smaller symbols) are inset.**



### 3.4 Carbon isotope ratios of organic matter

The carbon isotopic ratio of organic matter of *C. braarudii* is decreasing with increasing $CO_2$ level (from 200 to 500 ppmv)
and decreasing pH (from 8.29 to 7.89 pH units) ($r^2$ = 0.93, p < 0.001). This result is mirrored by a large $\delta^{13}C_{coccolith}$ increase
(+2.4‰ V-PDB) (Figure 5). Above 600 ppmv (below 7.89 pH units), $\delta^{13}C_{org}$ of *C. braarudii* are stable and close to the
values observed for the 200 ppmv/8.29 pH units condition (-30.77 ± 0.36‰ V-PDB) (Figure 7). $\delta^{13}C_{org}$ of *G. oceanica* do
not change with changes in $p$CO_2$ and pH (mean of -29.12 ± 0.41‰ V-PDB), similar to what was described for the changes
in $\delta^{13}C_{coccolith}$.


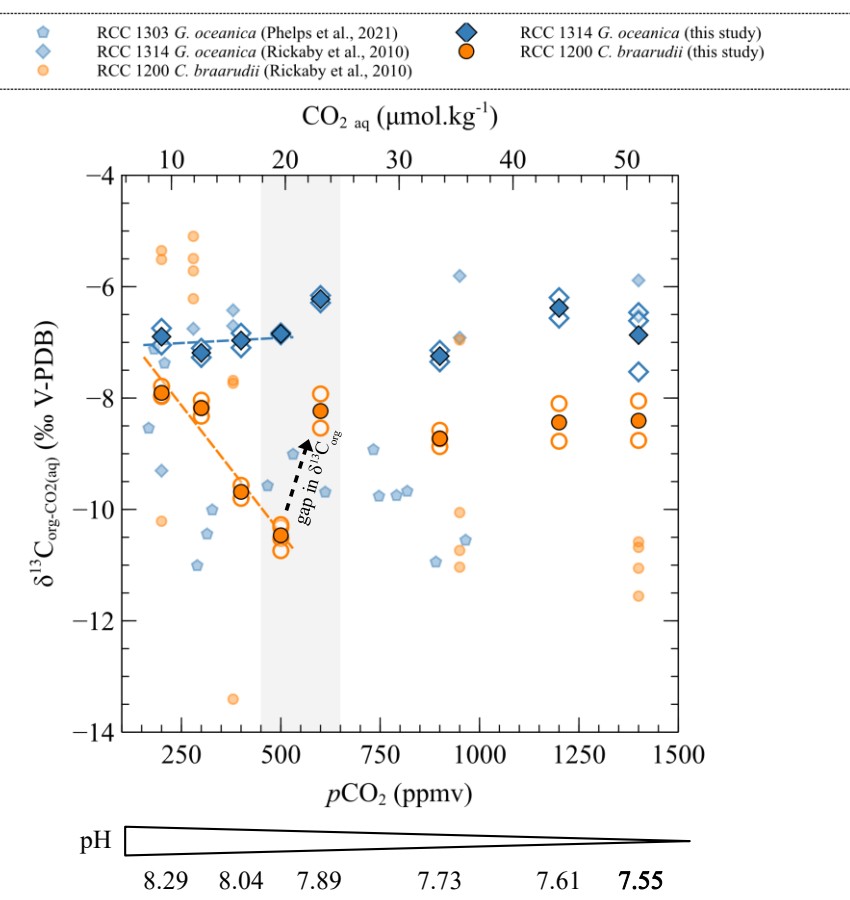

**Figure 7: Carbon isotopic ratios of organic matter compared to $\delta^{13}C_{CO2\,aq}$ of the strains studied with changes in $CO_2$ level and pH.
The empty points are all the replicate data and the filled points are the means. The grey band represents the carbonate chemistry
condition where a shift in the carbon and oxygen isotope ratio occurs. The sources of previously reported data (smaller symbols)**
**are inset.**



## 4 Discussion

The aim of our study is to quantify the environmental forcing exerted by $CO_2$ availability and ambient pH on the expression of vital effects of the coccolithophores with a view to develop and further palaeoenvironmental proxies. Our experimental results shed light onto the link between the environment, cellular growth and the efficiency of carbon fixation, as these parameters collectively control the expression of biologically-induced fractionation in organic and inorganic calcite. Published literature has revealed the role of carbon acquisition throughout the cell membrane and intracellular utilisation of carbon on the magnitude of the vital effect through laboratory and modelling studies (Laws et al., 1995; McClelland et al., 2017; Popp et al., 1998; Rau et al., 1996) Bolton and Stoll (2013) coined the concept of the demand-to-supply ratio to characterise the isotopic implication of carbon trafficking within the cells with large implication of the carbon isotope vital effect at different timescales. From these studies, it has emerged that the carbon isotopic composition of coccolith calcite was primarily controlled by the interplay of the inorganic carbon fixation in the coccolith vesicles and that of organic carbon in the chloroplast. At constant source of external carbon, the main driver dictating the $\delta^{13}C_{coccolith}$ is the amount and isotopic composition of organic matter produced through photosynthetic carbon fixation (POC/cell and $\delta^{13}C_{org}$ respectively) (McClelland et al., 2017).

As the foremost finding of our study, we show that *C. braarudii* exhibits a large $\delta^{13}C_{coccolith}$ increase (+2.4‰ V-PDB) mirrored by a $\delta^{13}C_{org}$ decrease (-2.6‰ V-PDB) from 200 ppmv and 8.29 pH units to 500 ppmv and 7.96 pH units (low $p$CO$_2$ and high pH conditions) (Figure 8). The correlation between $\delta^{13}C_{coccolith}$ and $\delta^{13}C_{org}$ of *C. braarudii* is not seen at high $p$CO$_2$ and low pH (600 to 1400 ppmv/7.89 to 7.55 pH units) (Figures 5 and 7). Based on this observation, we will treat separately the data obtained at low $p$CO$_2$/high pH from those obtained at high $p$CO$_2$/low pH in the following sections. It is also noteworthy that the small species *G. oceanica* and *E. huxleyi* do not show statistically-significant changes in $\delta^{13}C_{coccolith}$ nor $\delta^{13}C_{org}$ values coeval with changes in $CO_2$ level and pH (neither at low $p$CO$_2$ and high pH, nor over the whole interval).

### 4.1 The biogeochemical causes for the changes in $\delta^{13}C_{coccolith}$ of *C. braarudii* at low $p$CO$_2$ and high pH

Given that organic compounds are significantly $^{12}$C-enriched relative to $CO_2$ with typical $\delta^{13}C_{org}$ values around –25‰, the more organic matter produced, the isotopically heavier the residual intracellular carbon pool. As calcification derives from this latter pool, coccoliths produced by highly photosynthetic coccolithophore cells exhibit relatively higher $\delta^{13}C$ values. This phenomenon is particularly expressed in small cells such as *E. huxleyi* and *G. oceanica* that are characterised by high POC contents and the highest $\delta^{13}C$ values (Ziveri et al., 2003) (Figures 4 and 5). We can hypothesise that a change in the efficiency of POC production with changes in carbon availability can induce a change in $\delta^{13}C_{coccolith}$ values, as it is apparent for *C. braarudii* with +2.4‰ V-PDB shift between low and high $p$CO$_2$ conditions (Figure 5). There is no apparent change in POC, PIC/POC or growth rates with the increase in $\delta^{13}C$ at low $p$CO$_2$ and high pH (Figures 2 and 4), despite an increase in





chlorophyll a concentration within the cells between 200 and 500 ppmv (Figure A2). These observations exclude a control of the amount of organic matter produced on the isotopic signature of the internal pool. Furthermore, a change in the efficiency of POC production could not explain the coeval -2.6‰ and progressive decrease in $\delta^{13}C_{org}$ values within this low $p$CO$_2$

interval. Thus, another biogeochemical process has to be sought.

A second means to explain coeval changes in both $\delta^{13}C_{org}$ and $\delta^{13}C_{coccolith}$ can rely on a change in the isotopic composition of the carbon acquired by the cell (CO$_2$ vs. HCO$_3^-$). Carbon concentrating mechanisms (Giordano et al., 2005) can induce a shift in internal carbon pool to which a contribution by HCO$_3^-$ ions by active transport becomes significant (Bolton and Stoll, 2013). Under this circumstance, the internal carbon may exhibit higher $\delta^{13}C$ values, as there is a typical 9‰ equilibrium

fractionation between CO$_2$ and HCO$_3^-$ (Zeebe and Wolf-Gladrow, 2001). However, early biological work aiming at characterising inducible CCMs in phytoplankton has revealed the lack of such active strategies of carbon acquisition in the large and ancestral *Coccolithus* taxon. Furthermore, the build-up of the carbon pool with a proportion of HCO$_3^-$ would have led to higher $\delta^{13}C$ values in both organic and inorganic pool, which is not what is observed at lowest $p$CO$_2$ (conversely, the $\delta^{13}C_{coccolith}$ values are lower) (Figures 5 and 7). These two lines of evidence rule out the CCM hypothesis.

The magnitude of carbon isotope fractionation between the CO$_2$ substrate and the organic matter is not constant and can change with the amount of ambient CO$_2$ forming the ground of the $\varepsilon_{p-alk}$ proxy (Pagani, 2002; Popp et al., 1998). This modulation of the carbon kinetic fractionation by RuBisCO not only has consequences on the isotopic signature of the organic matter of which the compound-specific as alkenones, but also on the residual carbon pool, and ultimately on coccolith calcite.

Culture and wild coccolithophore data have revealed that the carbon isotope composition of the organic matter decreases with increasing CO$_2$ concentration (Bidigare et al., 1997). This biogeochemical control is compatible with the 2.6‰ decrease observed in $\delta^{13}C_{org}$ of *C. braarudii* between 200 and 500 ppmv (r² = 0.93, p < 0.001) (Figure 7). As the inorganic and organic carbon pool are linked at least from an isotopic perspective, we can interpret the +2.4‰ change in $\delta^{13}C_{coccolith}$ as the result of a modulation of kinetic fractionation of the organic matter via the increasing CO$_2$ availability (Figure 8).

As we do not observe any change in the growth rate nor in the POC per cell between low and high $p$CO$_2$ conditions, we suggest that the cells compensate for the decrease in carbon bioavailability in a different way. Indeed, the lower surface-to-volume ratio at 200 ppmv compared to 500 ppmv (cell diameter of 16.7 ±0.2 µm at 200 ppmv and 17.6 ±0.4 µm at 500 ppmv, in Fig. 3) may compensate for the lower supply with a lower demand to sustain growth rate (Bolton and Stoll, 2013; Rau et al., 1996).




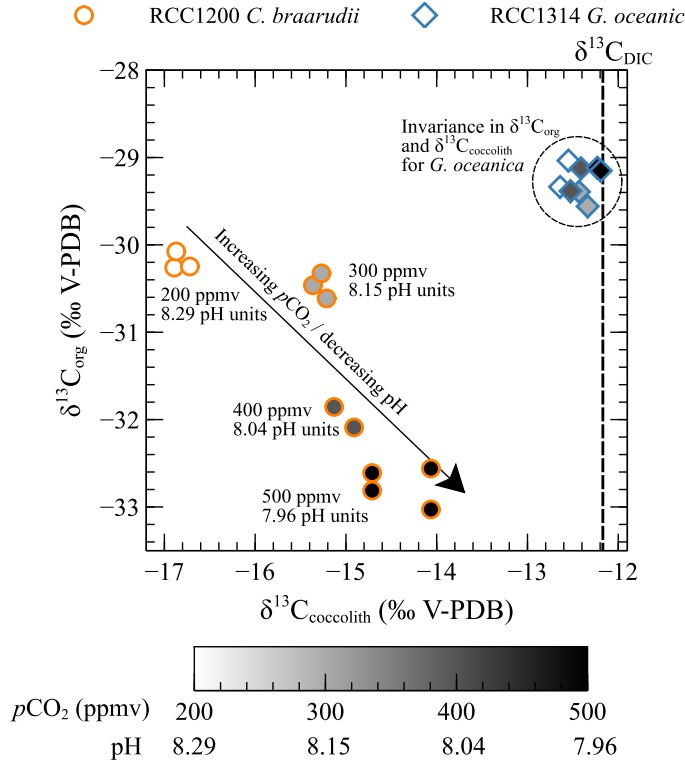

**Figure 8: Scatter plot showing $\delta^{13}C_{org}$ versus $\delta^{13}C_{coccolith}$ values with changes in $pCO_2$ and pH (circles: *Coccolithus braarudii*, RCC 1200; diamonds: *Gephyrocapsa oceanica*, RCC 1314). Grey levels correspond to growth conditions (white: 200 ppmv/8.29 pH units, light grey: 300 ppmv/8.15 pH units, dark grey: 400 ppmv/8.04 pH units, black: 500 ppmv/7.96 pH units).**


## 4.2 Abrupt change in the isotope biogeochemistry of *Coccolithus braarudii* in response to alleviated carbon limitation and enhanced proton concentration in the cell

The data for *C. braarudii*, and in particular the link between $CO_2$ availability and the isotopic composition of coccoliths and organic matter, cannot be explained by a uniform biogeochemical framework, as a gap is seen for *C. braarudii* between 500

and 600 ppmv (Figures 5 and 7). The data show that above 600 ppmv, there is no correlation between ambient $CO_2$ levels and $\delta^{13}C_{org}$. Furthermore, there is no covariation between $\delta^{13}C_{org}$ and $\delta^{13}C_{coccolith}$ values of *C. braarudii* at high $pCO_2$ and low pH, in contrast to the covariation observed at low $pCO_2$ and high pH. The latter point suggests that the organic-to-inorganic forcing that occurs at low $pCO_2$ levels no longer operates at high $pCO_2$ levels. The fact that there is no organic-to-inorganic forcing at high $pCO_2$ is presumably due to the alleviation of carbon limitation, or at least due to a lower carbon demand-to-

supply ratio (Bolton et al., 2012). Likewise, coccosphere sizes exhibit no discernible trend with $CO_2$ concentrations at high $CO_2$ levels (Figure 3), in contrast to the findings at low $CO_2$ levels, which were tentatively attributed to a way to enhance $CO_2$ influx to the cells. This biogeochemical feature would indicate that the forcing of $CO_2$ availability on the apparent





$^{13}C/^{12}C$ fractionation between the organic matter and calcite occurs only below 600 ppmv. Meanwhile, small species (*E. huxleyi* and *G. oceanica*) show unchanged isotopic values with the $p$CO$_2$/pH treatments. As explained in Rickaby et al.

(2010) and Hermoso et al. (2016), the high surface to volume ratio of the small cells induces no carbon limitation at low CO$_2$ level, hence no impact of the change of $p$CO$_2$ and pH on the carbon isotopic system.

Elevated $p$CO$_2$ conditions are accompanied by greater proton concentrations in the environment (more acidic conditions). Calcifying phytoplankton such as the Coccolithophores has to efflux the excess protons generated by calcification by an active process operated by transmembrane Hv channels (Kottmeier et al., 2022; Taylor et al., 2011). Previous biological

studies have demonstrated that the opening of Hv channels are affected by intracellular pH (Taylor et al., 2011). Hv channels are closed at low pH to counter the influx of protons into the cell from the ambient environment. The pH threshold for the closure of the Hv channels has been assessed to be within the range of 8.1 – 7.5 pH units (Kottmeier et al., 2022). Therefore, assuming that the threshold stands between 7.96 and 7.89 pH units, the closure of the Hv channels is a good candidate for the 500-600 ppmv gap apparent in our dataset (Figures 5, 6 and 7). The closure of the Hv channels can thus result in an

accumulation of H$^+$ ions within the cytosol, leading to an intense decline in intracellular pH (Kottmeier et al., 2022; Taylor et al., 2011). The release of protons by coccolithogenesis is even more influential on intracellular pH for species such as *C. braarudii*, which produces a large amount of PIC compared to smaller cells (Figure 4). The latter cells (*G. oceanica* and *E. huxleyi*) indeed do not exhibit such a gap in δ$^{13}$C$_{org}$ between 500 and 600 ppmv (small mean PIC values of 0.025 ±0.004 ng/cell and 0.006 ±0.001 ng/cell in the 600-1400 ppmv interval, respectively). Interestingly, the gap in *C. braarudii* δ$^{13}$C$_{org}$

between low and high CO$_2$ levels is not seen in δ$^{13}$C$_{coccolith}$. This observation could be the consequence of a decrease in the organic/inorganic carbon pool coupling when carbon supply is high, unlike bellow 500 ppmv when the small carbon pool imposes a high isotopic dependence between organic matter and calcite through Rayleigh distillation governed by the organic carbon fixation.

Around the same $p$CO$_2$/pH limit as presented for δ$^{13}$C$_{coccolith}$ of *C. braarudii*, we observe a change in δ$^{18}$O$_{coccolith}$ values. The

mean VE$^{18}$O value is –0.81 ±0.13‰ V-PDB at low $p$CO$_2$/high pH conditions, while the mean VE$^{18}$O value is –0.37 ±0.12‰ V-PDB at high $p$CO$_2$/low pH conditions (Figure 6). The gap in δ$^{18}$O$_{coccolith}$ can be explained by the change in pH. When the pH in the environment is low, it has been demonstrated that the $^{18}$O/$^{16}$O equilibration time between DIC species and H$_2$O is shorter compared to high pH conditions (Usdowski et al., 1991). The isotopic equilibration between δ$^{18}$O$_{DIC}$ and δ$^{18}$O$_{sw}$ due to the assimilation of isotopically heavy CO$_2$ into the cell is consequently more complete at low pH than at high pH. As a

consequence, δ$^{18}$O$_{DIC}$ (and consequently δ$^{18}$O$_{coccolith}$) values are closer to δ$^{18}$O$_{sw}$ at high $p$CO$_2$ and low pH (Figure 6). The δ$^{18}$O$_{coccolith}$ difference between high and low pH levels is consistent with a change in pH homeostasis strategy between 7.96 and 7.89 pH units that can be responsible for the δ$^{13}$C before and after 500/600 ppmv.





**4.3 Palaeoclimatic implications of the CO₂ proxy based on δ¹³C$_{coccolith}$**

**4.3.1 Δδ¹³C$_{small-large}$ evolution with changes in CO₂ levels and pH**

The increase in δ¹³C$_{coccolith}$ of *C. braarudii* with rising $p$CO₂ and decreasing pH shows potential as a proxy for reconstructing past carbonate chemistry, and consequently atmospheric $p$CO₂. However, utilizing absolute carbon vital effects of coccoliths for palaeo-$p$CO₂ reconstructions requires accurate knowledge of the δ¹³C$_{inorganic}$ (DIC) of past oceans, which remains challenging (Hermoso et al., 2020). To overcome this issue, previous studies (Bolton et al., 2012; Bolton and Stoll, 2013;

Godbillot et al., 2022; Hermoso et al., 2016b; McClelland et al., 2017; Tremblin et al., 2016) have proposed the use of the offset between the δ¹³C$_{coccolith}$ of small and large species, known as the differential vital effect (noted Δδ¹³C, Eq. (3)). Indeed, the δ¹³C$_{coccolith}$ of the small cells (Noelaerhabdaceae) is steady regardless of the $p$CO₂ levels due to their relatively large carbon pool, akin to *C. braarudii* under carbon replete conditions (at high $p$CO₂ and low pH). The coccolith-based palaeo-$p$CO₂ proxy is independent of the isotope ratio of ambient DIC, as both small and large coccoliths are produced by cells

growing in the same shallow water. In practical terms, Δδ¹³C$_{small-large}$ can be expressed as follows:

$$\Delta\delta^{13}C_{small-large} = \delta^{13}C_{small\ coccolith} - \delta^{13}C_{large\ coccolith} , \quad (3)$$

where δ¹³C is expressed in ‰ V-PDB. The small coccoliths used in this work are those of *G. oceanica* and the large coccoliths are those of *C. braarudii*.

Δδ¹³C and $p$CO₂ data highlight a linear relationship across the range of 200 to 500 ppmv (i.e., pH values between 8.29 and

7.96), that we previously explained by a modulation in the carbon isotope fractionation between CO₂ aq and the organic matter (Figure 9, Equation (4)). The uncertainties associated with the constants in Eq. (4) represent the standard deviation obtained with a linear regression model.

$$CO_{2\ aq}(\pm3.17) = -4.65(\pm0.72) \times \Delta\delta^{13}C_{G.\ oceanica-C.\ braarudii} + 26.90(\pm2.21) , \quad (4)$$

where CO₂ aq is expressed in µmol.kg⁻¹ and Δδ¹³C is expressed in ‰ V-PDB.




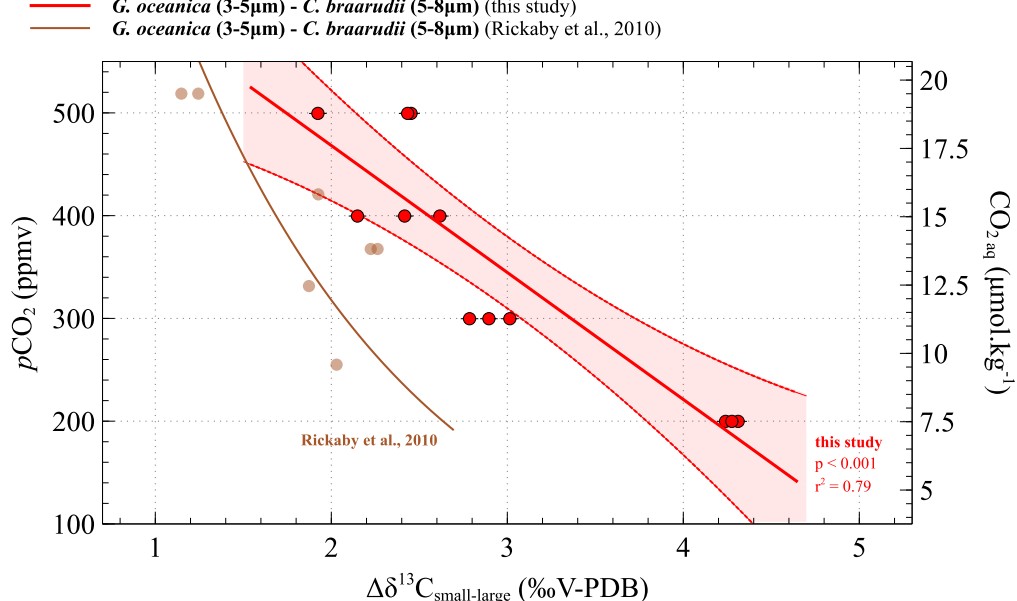

**Figure 9: Calibration between $CO_2$ and $\Delta\delta^{13}C_{small-large}$ obtained from culture experiments. The results of our study are shown in red (replicate data as dots, the linear regression made between 200 and 500 ppmv corresponds to the solid line and the confidence interval is shown with dotted lines). The brown curve is the equation recalculated from Rickaby et al. (2010) dataset. The relation encompassing the 200-1400 ppmv spectrum is shown in Figure A1.**

### 4.3.2 Palaeoclimate application of carbon isotope culture data

In this study, we refine a geological $CO_2$-sensitive probe by the use of the $\Delta\delta^{13}C_{small-large}$ isotopic offset. This probe forms the basis for a palaeo-$CO_2$ proxy transferable to sedimentary records from geological periods with low $CO_2$ levels, such as the Neogene and the Quaternary (Bolton and Stoll, 2013; Godbillot et al., 2022; Hermoso et al., 2020). Providing a new palaeo-$pCO_2$ proxy for those periods is of key interest, as it has been shown that alkenones are less sensitive to low and medium $CO_2$ level changes (Badger et al., 2019). Another strength of the $pCO_2$ proxy presented here is that it is based on coccolith calcite, which can be separated from other sedimentary components according to their size ranges by microfiltering and centrifuging protocols (Minoletti et al., 2008; Stoll and Ziveri, 2002; Zhang et al., 2018, 2021).

In addition, our study provides a biogeochemical explanation for $\Delta\delta^{13}C_{small-large}$ changes with $CO_2$ level and pH, thus supporting the reliability of these equations for palaeoclimate applications. We also demonstrate that the $\Delta\delta^{13}C_{small-large}$-$CO_2$ calibration can be extended to the whole interval studied, i.e., above 600 ppmv (Figure A1). Despite the fact that the link between $\Delta\delta^{13}C_{small-large}$ and $CO_2$ levels is less constrained and understood at high $CO_2$ level than at low $CO_2$ levels, the relationship is statistically supported on the whole interval ($r^2 = 0.80$, $p < 0.001$) (Figure A1). Thus, this calibration can potentially be used to reconstruct palaeo-$CO_2$ levels throughout the entire Cenozoic era.



## 5 Conclusion

The combined study of the impact of $CO_2$ concentration and pH on the fractionation of carbon and oxygen isotopes in coccolithophores (in their organic matter and calcite biominerals) provides an explanation of the cause of variations in $\Delta\delta^{13}C_{small-large}$. One of the major findings of this study is the coeval variation of $\delta^{13}C_{coccolith}$ and $\delta^{13}C_{org}$ in *Coccolithus*

*braarudii* with changes in $p$$CO_2$ between 200 and 500 ppmv and pH between 8.29 and 7.96. Combined with the fact that physiological parameters (growth rates, PIC, and POC) of *C. braarudii* remain unchanged despite changes in the availability of carbon, these results indicate that the cause of variations in $\delta^{13}C_{coccolith}$ is an environmental-driven change in the magnitude of the fractionation between ambient $CO_{2\ aq}$ and organic matter produced by *C. braarudii*. Above 500 ppmv, and for pH values below 7.96, greater carbon availability induces isotopic decoupling between $\delta^{13}C_{org}$ and $\delta^{13}C_{coccolith}$. On the other

hand, small species exhibit no change in $\delta^{13}C_{coccolith}$ or $\delta^{13}C_{org}$ in response to changes in $CO_2$ levels and pH. By comparing $\delta^{13}C_{coccolith}$ of small (*G. oceanica* or *E. huxleyi*) with those of large cells (*C. braarudii*), we have established an insightful $\Delta\delta^{13}C$-$CO_2$ transfer equation relevant for the Neogene and Quaternary timeslices.



**Appendices**

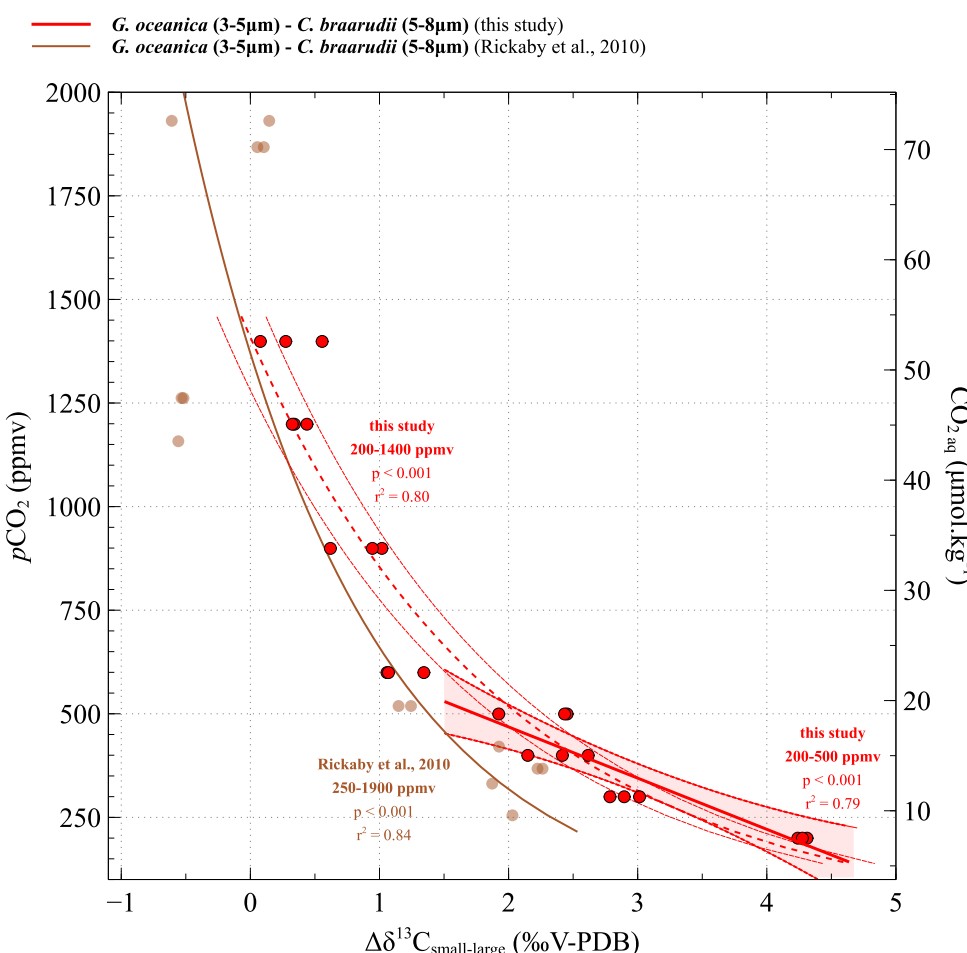

**Figure A1: Calibration between CO₂ concentrations and Δδ¹³C$_{small-large}$ obtained from culture experiments at low and high CO₂ levels. The results of our study are shown in red. Replicate data are illustrated with red dots. The linear regression made between 200 and 500 ppmv corresponds to the solid line and the confidence interval is shown with dotted lines. The dotted line shows the logarithmic regression made for the data on the whole interval (200 to 500 ppmv). The brown curve is the equation recalculated from Rickaby et al. (2010) dataset.**



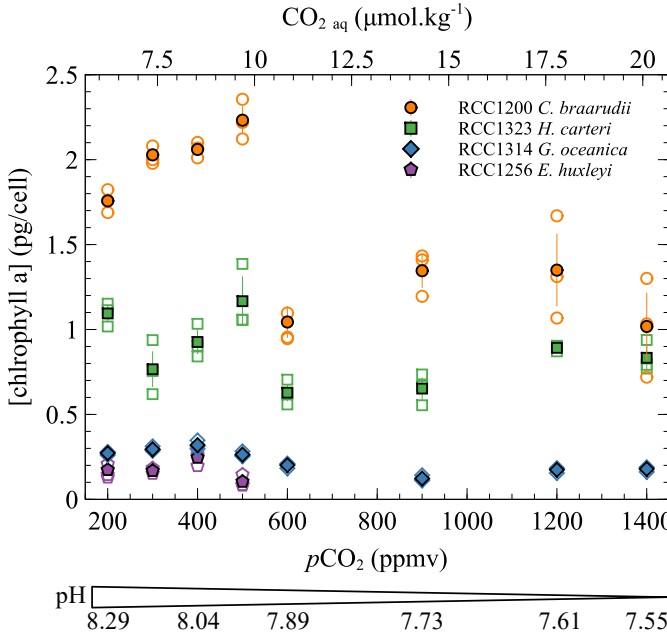

**Figure A2: Chlorophyll a concentration (pg/cell) as a function of CO₂ and pH. The empty points are all the replicate data and the filled points are the means (circles: _Coccolithus braarudii_, RCC 1200; diamonds: _Gephyrocapsa oceanica_, RCC 1314; squares: _Helicosphaera carteri_; pentagons: _Emiliania huxleyi_). Measurements of chlorophyll a concentration were conducted using** 505 **fluorometry according to the SOMLIT national protocol** (Yentsch and Menzel, 1963)**. A Turner Design Trilogy fluorometer was employed to measure the fluorescence of our samples before and after acidification. The samples were acidified with 10 µL of hydrochloric acid per ml of acetone extract and left in the dark for 2 minutes between the two measurements. The equation used to calculate the chlorophyll concentration is as follows (Lorenzen, 1967).**

**Data availability:**

All numerical data generated in this study are included in the Supplement.

**Supplement:**

The data related to this article will be available online upon publication at: https://doi.org/10.5281/zenodo.12187457

**Author contributions:**

This study was conceived by MH and FM. Experiments were undertaken by GLG, CG and MH. Measurements were done by GLG, CG, GD and VR. Data were analysed by GLG, MH and FM. The paper was written by GLG, FM and MH with inputs from the other authors.



**Competing interests:**

The authors declare that they have no conflict of interest.

**Acknowledgements:**

We thank L. Emmanuel and A. Guittet for help with the calcite isotopic analyses and O. Boudouma for help on the SEM at the laboratory ISTeP, Sorbonne University. We are also grateful to G. Reverdin, J. Demange, C. Waelbroeck and J. Fin for isotopic and DIC concentration analyses in the culture medium at LOCEAN, Sorbonne University. For help with isotopic analyses on organic matter, we thank M. Ader and G. Bardoux from IPGP. Our thanks also go to I. Probert at the biological

station of Roscoff for providing the strains. We acknowledge with thanks the financial support from the French *Agence Nationale de la Recherche* (ANR) – Project CARCLIM under reference ANR-17-CE01-0004-01 and from the Graduate school IFSEA under reference ANR-21-EXES-0011 (France 2030 programme). We also benefited financial support from the French *Centre National de la Recherche Scientifique* (CNRS-INSU) Project TOPCAPI within the TelluS-SYSTER programme.

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
