# Peer review of "Multispecies expression of coccolithophore vital effects with changing CO2 concentrations and pH in the laboratory with insights for reconstructing CO2 levels in geological history"

_EGUsphere, 2024_

## Author Response (AR1)

Dear Pr. Middelburg,

We are very grateful to Reviewer Hongrui Zhang for his insightful comments on our original manuscript. We are pleased to see that the reviewer appreciated the manuscript and recognized the value of the data, although he raised a number of points to address. As you will see in the point-by-point responses to his comments below, a revised manuscript can incorporate all the requested changes without altering the philosophy of the paper.

Dear Editor,

I have read the manuscript of "Multispecies expression of coccolithophore vital effects with changing CO2 concentrations and pH in the laboratory with insights for reconstructing CO2 levels in geological history" by Le Guevel et al.

In this work, they cultured four strains of coccolithophore under different CO2/pH conditions. The carbon and oxygen isotope fractionations of organic carbon and coccoliths were carefully measured. This new dataset could enrich our knowledge of the coccolithophore isotopic fractionations and is very valuable to future works in simulating coccolithophore isotope ratios and reconstructing the past pCO2. I believe that this work can be published after reorganizing the discussion part.

Here are my suggestions for their discussion:

[1] The organic carbon isotope fractionations of E. huxleyi and G. Oceanica did not show clear variations with CO2, even in high CO2 conditions. This point could be very attractive to many researchers using alkenone carbon isotope fractionations, but was not discussed in this draft.

We agree. However, we feel not ideally poised to tackle such a discussion (as we did not measure alkenone C-isotope compositions). We can add a sentence in the discussion (perhaps also in the conclusion) stating that the invariant nature of inorganic and bulk organic isotopic composition with our $p$CO$_2$ treatment would require closer examination by molecular geochemists.

[2] The concept of carbon limitation was mentioned throughout the whole paper. However, the authors did not try to calculate it. I think that it will improve the draft a lot if you plot the isotopic effect against the carbon limitation.

In previous published studies by the research team, we indeed approached the notion of carbon limitation by the (mu*volume)/(CO$_2$*surface) ratio (derived from Bidigare et al., 1997). The graphs are provided below. Figure R1 shows the change in the isotope ratios with varying CO$_2$ concentration.

In our study, one of the results is that (i) growth rates remain unchanged despite the imposed changes in CO$_2$ concentration and pH for all strains, and (ii) cell size remains unchanged

despite imposed changes in $CO_2$ concentration and pH for the small strains (namely *G. oceanica* and *E. huxleyi*). It thus turns out that the $CO_2$ concentration is the only parameter that dictates the carbon limitation here. Indeed, we can see that when the Bidigare index is calculated keeping $CO_2$ constant, carbon limitations of all strains are very close to each other (Figure R2). This is why we initially decided not to represent the isotopic data as a function of the carbon limitation, but only as a function of $CO_2$ level. This will be discussed in the revisions. The figures can be added to the manuscript as supplemental information.

[Figure]

*Figure R1: Carbon (A) and Oxygen (B) isotopic effect with changes in the carbon limitation. Carbon limitation is calculated as the product of growth rate (μ, in day$^{-1}$) and coccosphere volume (V, in μm$^3$) divided by the product of $CO_2$ concentration ($CO_2$, in μmol.kg$^{-1}$) and coccosphere surface (S, in μm$^2$), according to Bidigare et al., 1997.*

[Figure]

*Figure R2: Carbon (A) and Oxygen (B) isotopic effect with changes in the carbon limitation calculated as for Figure R1 with the difference that the $CO_2$ concentration is a constant (25.82 μmol.kg$^{-1}$, mean $CO_2$ concentration of the medium conditions in this study).*

*Bidigare, R. et al. (1997) Consistent fractionation of 13C in nature and in the laboratory: Growth-rate effects in some haptophyte algae, Global biogeochemical cycles, 11(2), pp. 279–292.*

[3] For the fractionations observed in the C. Braarudii (section 4.1), the authors explained it as "moderation of kinetic fractionation of the organic matter (line 384)". I don't understand what this is. Do the authors mean that the fractionations of RubisCO or the Rayleigh fractionation in chloroplast? Please don't use this kind of misty description after four paragraph discussions.

This part was maybe not clear enough. We indeed refer to the kinetic fractionation by the fixation of C by the enzyme RusbiCO. This was introduced in previous sentence lines 375-377. To avoid any ambiguity, we will add this again at the beginning of section 4.1.

[4] I feel that some discussion in  Section 4.2 answered the questions in Section 4.1. So, I would suggest reorganizing these two sections and short the length.

We intended to make our demonstration apparent with an emphasis on the low (<600 ppmv) $CO_2$ range. To this aim, we first enunciate the possible explanations for the isotopic changes observed below 600 ppmv and we provide a discussion of each. Then, we discuss the possible explanations for the abrupt change for *C. braarudii* $\delta^{13}C_{coccolith}$ around 600 ppmv. For each aspect of Section 4.1 that is left unresolved, we propose to add a comment as "This point will be clarified in Section 4.2" or an equivalent sentence.

[5] In the section 4.3, the authors are too optimistic about this proxy. It is better to emphasize what is the changing part of future work. For example, how to use it when there is no C. Braarudii in tropical sediment?

*Coccolithus braarudii* is not (anymore) a species thriving at low latitudes. Indeed, coccoliths of the *Coccolithus pelagicus* group are absent in warm, low latitude waters since the Plio-Pleistocene boundary (Sato et al. 2004, MarMic). This work shows that before the onset of this palaeogeographic feature, *C. pelagicus* coccoliths were present in all sediments from the equator to the mid latitudes (50°N) in the Atlantic and Pacific oceans.

Our aim is to develop a paleoenvironmental proxy of $CO_2$ concentration applicable to deep time archives where *C pelagicus* is present and sometimes abundant at all latitudes making this species a relevant target for our culture experiment.

*Tokiyuki Sato, Shiho Yuguchi, Toshiaki Takayama, Koji Kameo (2004) Drastic change in the geographical distribution of the cold-water nannofossil Coccolithus pelagicus (Wallich) Schiller at 2.74 Ma in the late Pliocene, with special reference to glaciation in the Arctic Ocean. Marine Micropaleontology 52, 1–4, pp 181-19. https://doi.org/10.1016/j.marmicro.2004.05.003*

Can we have a good calibration of pCO2 by the modern surface sediment?

Such a work has been attempted (Hermoso et al., 2015) and somehow "failed" to establish a $\delta^{13}C/pCO_2$ relationship for Noelaerhabdaceae coccoliths and possible reasons are discussed in the paper.

*Michaël Hermoso, Yaël Candelier, Thomas J. Browning, Fabrice Minoletti, (2015) Environmental control of the isotopic composition of subfossil coccolith calcite: Are laboratory culture data transferable to the natural environment?, GeoResJ, Volume 7, Pages 35-42.* https://doi.org/10.1016/j.grj.2015.05.002

There are two calibrations in Figure 9, which one is better? If we do another cluster of coccolithophore isotopic culture, will we get another different calibration? I am not saying that all there questions should be well answered, but please at least inform the reader what is the limitation of the current work and which direction we should go.

It is noteworthy that the two calibrations match pretty well, at least to first order. Of course colleagues are free to choose whichever published calibration for their work. Our study presents the advantage of bearing more measurements at low $pCO_2$ compared with the work of Rickaby et al., 2010.This was the rationale behind our culture work here.

One research avenue is the covariation of $CO_2$/pH levels crossed with temperature changes. This latter point can be added in the revised draft.

*Rickaby, R.E.M., Henderiks, J. and Young, J.N. (2010) 'Perturbing phytoplankton: Response and isotopic fractionation with changing carbonate chemistry in two coccolithophore species', Climate of the Past, 6(6), pp. 771–785. https://doi.org/10.5194/cp-6-771-2010*

Some detailed suggestions are blow:

Line 18: remove biominerals.

Coccolith biominerals can be changed to coccoliths

Line 19: could you please replace the "large-scale" by another word? This is not very clear for me. Too many "of" in one sentence.

We can modify the sentence to: "We have undertaken culture experiments of four coccolithophores strains with various sizes and growth rates, grown under eight $CO_2$/pH conditions typifying the $CO_2$ evolution of the Cenozoic Era".

Line  23: What do you mean by "on either side"

We mean below and above 600 ppmV. We can use "on both sides of the 600ppmV condition" to make the sentence clearer.

Line 30: I believe that we have known this framework for more than 10 years, since the publishment of Rickaby 2010 CP. So, please don't use the term "new" here.

We will delete new.

Line 61: I guess what you want to express here is "the diffusion of CO2 is governed primarily by the gradient of CO2 between inner cell and seawater, but also by the specific condition of cell boundary layer, such as the cell size..."

When it comes to Fick's Law, cell size of the exchange surface (i.e. the cell membrane), and the concentration gradient on either side of the membrane are taken into account (Reinfelder, 2011; Wolf-Gladrow and Riebesell, 1997).

*Reinfelder, J.R. (2011) Carbon concentrating mechanisms in eukaryotic marine phytoplankton, Annual Review of Marine Science, 3, pp. 291–315. https://doi.org/10.1146/annurev-marine-120709-142720*

*Wolf-Gladrow, D. and Riebesell, U. (1997) Diffusion and reactions in the vicinity of plankton: A refined model for inorganic carbon transport, Marine Chemistry, 59(1–2), pp. 17–34. https://doi.org/10.1016/S0304-4203(97)00069-8*

Line 81: I feel that a explanation is missing here: Why does this approach still need to be constrained, considering so many experiments well fit with each other in the past ten years?

The community of palaeoceanographers needs a consolidated calibration that can be used for any temperature and oceanic species. This implies the implementation of culture campaigns as that present in our paper. Such a work can only be incremental and need a lot of investigation regarding the multiple environmental parameters aside $CO_2$/pH.

Line 107: Please remove this sentence into Section 2.3 as the first one of that paragraph.

We thank the Referee for this, as this will indeed make the paragraph clearer.

Line 126: What is the target cell number?

In laboratory cultures of this nature, there is a trade-off between the need to keep the biomass (carbon utilisation) low and the need to generate enough material for further analyses. It is more a matter of mass of the culture residues than cell density (although these parameters are linked + culture volume and growth dynamics). The planned analysis (PIC/POC, $\delta^{13}C$ and $\delta^{18}O$ of calcite, $\delta^{13}C$ of organic matter, chlorophyll a concentration) require a minimal mass of around 50 mg for *C. braarudii* and *H. carteri*, around 10 mg for *G. oceanica* and around 2 mg for *E. huxelyi*. Our approach was thus to undertake very dilute culture batches where the carbon consumption was kept low (<5 %). To this aim, we adopted a semi-continous batch strategy by which the culture medium was refreshed every two-four days (depending on the growth dynamics and cell size). The higher cell concentration in the culture flasks were 8 000 cell/ml for *C. braarudii* and for *H. carteri* and 20 000 cell/ml for *E. huxleyi* and for *G. oceanica*. This is in accordance with the guidelines provided by Riebesell et al., 2011.

*Riebesell, U., Fabry, V. J., Hansson, L. and Gattuso, J.-P. (eds) (2011) Guide to best practices for ocean acidification research and data reporting. [reprinted edition including erratum]. Luxembourg, Publications Office of the European Union, 258pp. (EUR 24872 EN). DOI 10.2777/66906*

Line 129: What is the cell concentration as a diluted culture? Here I am not trying to question your work, but please give a clear definition to make it easier for other groups to repeat your work.

See above.

Line 130: 14/10 day-night cyclicity.

We can change the manuscript to "14/10 day-night cycles"

Line 149: I would recommend measuring the carbon isotope ratios of culture media in all experiments for your future work. I encountered a slight fractionation during the sterilization process.

We welcome with thanks this advice. Here, the culture media were never heated for sterilisation but 0.22-μm-filtered for this reason.

Line 216: I find that the growth rates of E. huxleyi varied with the CO2.

We agree that a trend can be observed at first glance by looking at the evolution of the mean values of the growth rate of *E. huxleyi* as a function of $CO_2$ concentration. However, the linear regression between replicate growth rates and $CO_2$ concentration below the 600 ppmv threshold gives a p-value of 0.05226 and a $r^2$ of 0.41. Thus, the relationship between growth rate and $CO_2$ is not statistically-significant.

Line 231: Do you try to say "The evolution of this parameter with CO2/pH varies among different strains."? The current sentence means that the size pattern among strains changes from low CO2 to high CO2.

We thank the Referee for this and we will modify the sentence accordingly.

Line 330: figure 7. The isotopic notation is wrong. When you describe the isotopic difference, please use either ε or Δ.

We cannot use Δ as this notation is already used for the isotopic offsets of coccoliths of different species in this work. We can use epsilon however, this is correct. We will change this in the final manuscript for each occurrence.

Line 342: unclear. Please rephrase it.

We can change the sentence to: Based on these studies, it appears that the carbon isotopic composition of coccolith calcite was primarily controlled by the interplay between inorganic carbon fixation in the coccolith vesicle and organic carbon fixation in the chloroplast.

Line 351: Though the background values of culture media carbon isotope ratios were the same among different experiments in your work, it is better to use fractionations when we discuss the isotopic effects, instead of the delta notation.

We will change this in the final manuscript for each occurrence.

Line 351: I don't find the results of organic carbon isotope fractionation in the E. huxleyi experiment.

We did not generate these data unfortunately. This is due to a limited availability of the instrument and fundings. We will add in the sentence in the data availability sections stating that the culture residues can be obtained upon demand.

Line 372: Reference for no CCM in Coccolithus.

Previous biogeochemical work has revealed that *Cocolithus* has limited extra- and transmembrane ability to express CCM and, as a consequence, the build-up of its internal carbon pool relies on passive diffusion of $CO_2$. We will add the reference to this seminal work of the revisions.

Nimer N. A., Ling M. X., Brownlee C. and Merrett M. J. (1999 .Inorganic carbon limitation, exofacial carbonic anhydrase activity, and plasma membrane redox activity in marine phytoplankton species. J. Phycol. 1205, 1200–1205. https://onlinelibrary.wiley.com/doi/10.1046/j.1529-8817.1999.3561200.x

Line 375-379: I find that these two sentences do not have good connections with context.

We suggest moving the sentence line 380 "*Culture and wild coccolithophore data have revealed that the carbon isotope composition of the organic matter decreases with increasing CO₂ concentration (Bidigare et al., 1997).*" before the sentences 375-379. This will provide a better context to the concept being discussed herein.

Line 408: Could you please separate the discussions of different species? I am really confused why I suddenly see the discussions on smaller cells without any translations, when the title of this section is mainly about C. Braarudii.

The logical of the discussion requires comparing large and small cells. We can however modify the heading of the 4.2 Section to make this apparent, as we can understand that the reader may expect to find only discussion on *Coccolithus* here.

4.2 Change in the isotope biogeochemistry of Coccolithophore cells in response to alleviated carbon limitation and enhanced proton concentration in the cell (*Coccolithus braarudii* versus smaller species).

Line 433: A discussion of Carbonic Anhydrase is missing here. If the coccolithophores express a lot of CA, which could accelerate the equilibrium by more than hundreds of times, the pH won't have a significant effect on the oxygen isotope.

This is correct. All microalgae cells indeed possess internal Carbonic Anhydrase (Paneth and O'Leary, 1985). We can add a sentence reminding this point to the reader. We don't know the kinetic of isotopic re*equilibration of the internal carbon pool within the cell compartments.

*Paneth P. and O'Leary M. H. (1985) Carbon isotope effect on dehydration of bicarbonate ion catalyzed by carbonic anhydrase. Biochemistry 24, 5143–5147.*

Line 453: Which calibration should the readers use, and why?

We guess that the Reviewer refers to Equation (3) and (4)? Eq. 3 is not a calibration but only the definition of the $\Delta^{13}C$ notation used in our work. The calibration appears in Eq 4.

Line 468: "we refine a geological CO2-sensitive probe". I feel that this proxy was not refined after knowing the new calibration is different with the Rickaby 2010 one...

As stated above, the two calibrations are not so different, especially given the fact that the culture methods and other parameters aside $CO_2$/pH were not the same (strains, temperature…).

We can change 'refine' for 'further develop'.

Hope my suggestions help to improve this draft.

Yes, they did, thank you again!

Hongrui Zhang (Tongji University)

Review of Le Guevel et al., 2024

Dear Pr. Middelburg,

Although the Reviewer did not really like our paper and tried to sink it through his/her report, we will take this opportunity to improve the manuscript as some of the comments are constructive - and we wish to thank the Referee for that. This Referee would have not had the same approach of the paper if she/he were the lead author. Our purpose here is the built-up of an empirical calibration transferable to the sedimentary record. The data are available to the colleague and he/she is more than welcome to use them with a different viewpoint.

Although we would welcome criticisms as there is always room for improvement, there are several statements about the methods and the care with which we undertook our cultures (or lack thereof) that do not appear to be justified.

You will read our responses after his/her general/specific comments below.

This paper presents experiments that aim to calibrate the isotopic vital effects in coccolithophore calcite with changing pCO2 such that the calibration may be used for isotopic values of different species in sediments to derive pCO2 values. I am afraid that overall, I found the writing of the paper was towards this preconceived aim, rather than truly exploring the nuance in the data and the lack of experimental details and constraint/measurement on the carbonate chemistry of the media almost renders the data meaningless at least for this purpose. In the first instance, the carbonate chemistry of the experiments does not replicate changes on geological timescales whereby alkalinity changes in concert with DIC and pCO2 due to the weathering feedback. The calibration merely changes the pH of the seawater to change pCO2 which is only relevant on timescales of <10kyrs. So the aim of the paper is undermined by the concept and execution of the experiments.

Unfortunately, it is impossible to precisely reproduce the changes in the carbonate system over geological time scales. Nevertheless, we try to come as close as possible to the inferred changes in this study by choosing $p$CO$_2$/pH couples compatible with the state of the oceans in (close) geological history. Furthermore, we have adjusted $p$CO$_2$ by adding NaHCO$_3$, as explained in Section 2.2. It was not the pH that changed $p$CO$_2$ of our culture media. See the Table 1:

| Culture conditions | pH (total scale) | TA (µmol/kg) | DIC (µmol/kg) | CO₂ (µmol/kg) | HCO₃⁻ (µmol/kg) | CO₃²⁻ (µmol/kg) | pCO₂ (ppmv) |
|---|---|---|---|---|---|---|---|
| | adjusted/ measured | calculated (CO2sys) | Fixed with NaHCO₃ | calculated (CO2sys) | calculated (CO2sys) | calculated (CO2sys) | Initial target |
| 1 | 8.29 | 2256.0 | 1913.6 | 7.51 | 1669.3 | 236.9 | 200 |
| 2 | 8.15 | 2275.0 | 2011.6 | 11.27 | 1813.9 | 186.5 | 300 |
| 3 | 8.04 | 2248.7 | 2042.2 | 15.03 | 1877.4 | 149.8 | 400 |
| 4 | 7.96 | 2272.4 | 2100.3 | 18.78 | 1951.9 | 129.6 | 500 |
| 5 | 7.89 | 2272.2 | 2128.8 | 22.54 | 1993.6 | 112.6 | 600 |
| 6 | 7.73 | 2269.0 | 2183.6 | 33.81 | 2068.9 | 80.9 | 900 |
| 7 | 7.61 | 2246.3 | 2199.7 | 45.08 | 2092.6 | 62.0 | 1200 |
| 8 | 7.55 | 2262.2 | 2233.8 | 52.59 | 2126.3 | 54.9 | 1400 |

Unfortunately the write up of the experiments misses out crucial pieces of information to be able to assess the quality and importance of the data. There is no actual measurement of the carbonate system parameters in the media. As far as I can tell from the Tables presented, only pH was measured, and there was no check or actual measurement on any other carbonate system parameters- they all seem to have been inferred by calculation from CO2sys

Once again, DIC and pH were measured. Therefore we can state that the carbonate chemistry was constrained since only two parameters are sufficient (*Zeebe and Wolf-Gladrow, 2001*).

pH was indeed measured for each medium produced. The concentration of carbon present in the medium is controlled by the amount of $NaHCO_3$ powder added during the preparation of artificial seawaters (see Section 2.2). The measurements of the amount of $NaHCO_3$ powder were made as precisely as possible, using a microbalance, with great care. It is the same batch of $NaHCO_3$ powder and method that was used in Hermoso, Chan and Rickaby (2016).

The DIC concentration induced by the addition of each amount of NaHCO3 was verified using an Isotope and Gas Concentration Analyzer Picarro G2131-i coupled with an Apollo SciTech DIC-$\delta^{13}$C analyser AS-D1 at LOCEAN laboratory (Sorbonne University). See paragraph 2.4 of Material and methods in the manuscript.

*Zeebe, R. and Wolf-Gladrow, D. (2001): CO2 in Seawater: Equilibrium, Kinetics, Isotopes , Elsevier Oceanography Book Series, 65, 346 pp*

*Hermoso, M., Chan, I.Z.X., et al. (2016) 'Vanishing coccolith vital effects with alleviated carbon limitation', Biogeosciences, 13(1), pp. 301–312. Available at: https://doi.org/10.5194/bg-13-301-2016*

I can see that they added NaHCO3 to achieve a certain concentration of DIC but what weights were added?

| Condition | $pCO_2$ (ppmv) | pH | NaHCO$_3$ added (mg/Lsw) |
|-----------|----------------|------|--------------------------|
| 1 | 200 | 8,29 | 164,68 |
| 2 | 300 | 8,15 | 171,94 |
| 3 | 400 | 8,04 | 176,59 |
| 4 | 500 | 7,96 | 179,9 |
| 5 | 600 | 7,89 | 182,41 |
| 6 | 900 | 7,73 | 187,47 |
| 7 | 1200 | 7,61 | 190,74 |
| 8 | 1400 | 7,55 | 192,44 |

This table will be add to the manuscript.

How carefully was this measurement made in terms of volume of media etc? In my opinion, this suggests that there is actually no control on the carbonate chemistry here as you need to measure two carbonate system parameters to be able to infer the other parameters. Unless this was oversight in the write up, none of the other carbonate system parameters are really known and the authors can only make plots of their data versus pH and can say nothing about pCO2.

Relative to the latter sentence, we strongly disagree with that statement that appears to be unjustified (see our responses above).

Furthermore, the volume of seawater produced was always higher than 5 liters to avoid inducing high errors when measuring too small masses of NaHCO$_3$. The masses of NaHCO$_3$ added for each target condition are referenced in the table above, which will be added to the manuscript.

We reiterate here: We have measured DIC and pH for each initial batch (and we know how much NaHCO$_3$ was added).

Furthermore, it does not state at what point in the experiment were the samples for media isotopic analysis taken? This is key for knowledge about drift in the isotopic systems.

The isotope ratios of the culture media are those at the start of the experiments, when the microalgae were inoculated. For the drift, see our Response to the other Reviewer (re line 129).

Second, due to the lack of measurement of the carbonate chemistry in the experiment, there is no knowledge about drift in the carbonate system and isotopes merely saying that large volumes were undertaken to keep the cultures dilute. But according to the best practices in ocean acidification work, the cultures should never use up more than 10% of the DIC? This is unknown in these experiments and the cultures may have driven significant isotopic drift in the media (https://www.iaea.org/sites/default/files/18/06/oa-guide-to-best-practices.pdf).

This remark closely relates to the one before. We are aware of this work and the reservoir effect. The research team has always endeavored to conform to these recommendations for more than ten years now. We were the first to claim that drifts in the carbon (both elemental and isotopic) systems can lead to abnormally elevated $\delta^{13}C$ values of coccolith calcite (Hermoso, 2014). We usually aim to maintain very diluted _AND_ implement semi-continuous batches for this reason although this is time consuming.

_Hermoso, M (2014). Coccolith-Derived Isotopic Proxies in Palaeoceanography: Where Geologists Need Biologists,Cryptogamie, Algologie 35(4), 323-351. https://doi.org/10.7872/crya.v35.iss4.2014.323_

For the organic C isotopes, there is no mention of precombustion of the GFF filters to remove adhering organics from the manufacture process so it is not clear whether these data are true or may suffer from contaminating blank issues? It states also that the organic C was scratched from the superficial of the filter, decalcified and then rinsed? Rinsed with what? Was this water carbon and carbonate free? These details are essential to be able to see if the trends in isotopes have any meaning at all.

The GFF filters were burnt at 400 °C for 3 hours to remove all traces of organic matter. We rinsed the culture residues with neutralized demineralized (Millipore) water (adjusted to pH 7 by adding sodium hydroxide). We will specify this in the revised manuscript.

There seems to be a rather fluctuating use of variable or not scales to axes on all of the graphs depending on the subjective choice of what is being shown. For the growth rate graphs, all are on different scales, but for the PIC/POC axes, they are all on the same scale. I am not sure I understand the reason for a 15% variance in the growth rate to show that the growth rates have no trend. Certainly the E. hux data look as though there is some kind of increase in growth rate with declining pH (increasing CO2 availability if that inference is possible). So why the value of 15% - it looks greater than the variance in measurements for any on point so is it justifiable.

All the graphs of growth rate are on the same scale for the absolute value (axis on the left), as well as for PIC:POC. We have chosen to represent all data on graphs with the same axes to facilitate comparisons between species. The 15 % interval chosen here (representing the scatters between replicates) is indeed arbitrary, and is only used to show that there is no trend.

The second reason to ask this, is on the Figure of PIC/POC for E. hux versus pH, there looks to be a decline in PIC/POC (as you might expect given the original paper by Riebesell et al., 2000) but the axes means the data are so compressed that it is hard to see. This suggests to me, that the authors are trying to show simplicity in the data to get towards the calibration rather than exploring the true message in the data. Why does the PIC/POC change or not change in E. hux with pH? This goes against most of what is known of this species.

The question about the growth rate of *E. huxleyi* is discussed in the responses for reviewer 1. The apparent trend is not statistically-supported.

It is unfortunate timing for this submission since the discussion also does not integrate recent results which have demonstrated an understanding of the biological mechanisms behind the coccolith vital effects (Chauhan and Rickaby, GCA, 2024) and does not discuss the comparators or dissimilarities with this new dataset.

The culture growth methodology used by Chauchan and Rickaby, 2024, and the strains studied - with the exception of *G. oceanica* - are not the same as ours, making a direct comparison not straightforward. The cultures of Chauchan and Rickaby, 2024 were grown at 17 °C under a light intensity between 55 and 80 $\mu mol.m^{-2}.s^{-1}$ during the day phase, whereas we grew the algae at 15 °C under a light intensity of 150 $\mu mol.photons.m^{-2}.s^{-1}$ during the day phase (twice as high).

The $\delta^{13}C$ of *C. braarudii* are relatively similar to the results of our work, albeit with slightly lower absolute values in Chauchan and Rickaby's work.

We must note that the $\delta^{13}C$ values observed by Chauchan and Rickaby for *G. oceanica* and for *E. huxleyi* are very negative for $CO_{2\ aq}$ concentrations greater than 20 $\mu mol/kg$ ($\delta^{13}C_{coccolith}$-$\delta^{13}C_{DIC}$ values between -2.5 and -8 ‰V-PDB), which is very surprising in comparison with the previous studies published in literature (Rickaby et al., 2010, Hermoso et al., 2016, McCLeland et al., 2017).

Moreover, the vital oxygen effects measured by Chauchan and Rickaby 2024 for *G. oceanica* makes this taxon belong to the 'light group' at high $CO_2$ concentrations - a feature that requires comments.

We will add some of these observations and comparisons in the results and in the discussion of our article.

Anyhow, this work deserves to be mentioned in the introducing presentation (state-of-the-art) of the paper, and it will be.

*Rickaby, R.E.M., Henderiks, J. and Young, J.N. (2010) 'Perturbing phytoplankton: Response and isotopic fractionation with changing carbonate chemistry in two coccolithophore species', Climate of the Past, 6(6), pp. 771–785. https://doi.org/10.5194/cp-6-771-2010*

*Hermoso, M., Chan, I.Z.X., et al. (2016) 'Vanishing coccolith vital effects with alleviated carbon limitation', Biogeosciences, 13(1), pp. 301–312. https://doi.org/10.5194/bg-13-301-2016*

*McClelland, H.L.O. et al. (2017) 'The origin of carbon isotope vital effects in coccolith calcite', Nature Communications, 8(May 2016). https://doi.org/10.1038/ncomms14511*

*Chauhan, N. Rickaby, R.E.M. (2024) Size-dependent dynamics of the internal carbon pool drive isotopic vital effects in calcifying phytoplankton, Geochimica et Cosmochimica Acta, Volume 373, Pages 35-51, https://doi.org/10.1016/j.gca.2024.03.033*

The manuscript fails to use the term G. huxleyi for this species which has now been suggested extensively and formally in the literature (see Bendif et al., 2023; Archontikis et al., 2023) due to the phylogeny of the organism and should either uptake the species terminology or explain why the former is used.

The name of strain RCC1256 is still *E. huxleyi*, according to the Roscoff biological station website that provided the strain (https://roscoff-culture-collection.org/rcc-strain-details/1256). Nannotax, the reference in calcareous microfossil taxonomy, still uses the name *Emiliania huxleyi* for this species (https://www.mikrotax.org/Nannotax3/cenozoic/Emiliania_huxleyi). To our best knowledge, the genus *Emiliania* has not been formally amended. We have therefore chosen to retain this name.

It is not clear to me that the authors have calculated the isotopic composition of the calcite correctly, as they have not accounted for the impact of pH particularly on the O isotopes. They do not reference Romanek et al., 1992 for C isotopes, Kim and O'Neill (1997) and Watkins et al., (2013) all of whom propose inorganic equations. Furthermore, particularly in experiments manipulating pH, Zeebe (1999) accounts for the $\delta18O$ of the precipitating calcite becoming lighter (depleted) with increasing pH as a consequence of the increasing proportion of the isotopically depleted $CO3^{2-}$. It is also not clear what corrections they have done to previously published data to get them onto the same axes. These should be detailed in the manuscript.

Indeed, pH has an impact on the $\delta^{18}O$ of the inorganic reference for certain pH ranges (as $CO_2$, $HCO_3^-$ and $CO_3^{2-}$ have distinct isotopic compositions). Our pH variations would only induce a very small variation in $\delta^{18}O$. All the data are expressed as differences in isotopic ratio. All the data used from the literature have been calculated as expressed on the axis of the graphs of our preprint.

I am afraid that the writing is quite sloppy throughout with sometimes totally opaque sentences: (see line 22 why should the fractionation into organic matter affect the lith?

We do not understand as line 22 is not related to this concept being referred to here. The manuscript will benefit from language editing by a native speaker.

see line 45: why should sensitivity <270 ppm have any implication for pCO2 estimates > 270 ppm?

Because the calibration curve follows a linear function, if the slope changes, we overestimate on one side and underestimate on the other.

and then just plain wrong scientifically (line 354 "organic compounds are 12C enriched") shows a plain lack of understanding of isotopic fractionation.

Organic compounds / matter always exhibit lower $\delta^{13}C$ relative to inorganic (calcite) compounds, they are thus relatively $^{12}C$-enriched. We do not understand what is wrong here, apart maybe from the English writing.

I am very sorry to say, as I was excited to read this manuscript, but I do not think that this manuscript should be published, at least in its current state.

---

## Author Response (AR2)

Dear Pr. Middelburg,

We thank again Reviewer Hongrui Zhang for reviewing the new version of our manuscript and are pleased that he is satisfied with the answers and suggested changes.

We would like to sincerely thank Reviewer #3 for taking the time to read and comment on our manuscript. We are pleased to hear that they found the data interesting and appreciated the way it was analyzed. We also appreciate their constructive comments (highlighted in orange) and address them point by point below (in black). The corresponding changes to the manuscript, primarily additions, are shown in blue.

*I could not find explanation how the pH was actually measured. Only the mentioning that it was on the total scale. Did you use an electrode with TRIS seawater buffers for calibration or spectrophotometry? Also, what is the uncertainty of this measurement and how would it propagate for calculated pCO2?*

We did not explicitly mention how the pH was measured in the original manuscript, and we thank the referee for pointing out this omission. The pH values were determined using electrometric measurements with a Mettler Toledo pH InLab® Routine pro-ISM electrode. Calibration was performed using a TRIS buffer solution (T34, Dickson Lab, see: CO2crms.ucsd.edu). The uncertainty of pH measurements with this electrode is estimated at 0.03 pH units (Meinrath and Spitzer, 2000). When propagated to the $p\text{CO}_2$ estimates using the CO2SYS spreadsheet, this uncertainty results in an error margin of ±15.5 ppmv for the $p\text{CO}_2$ of culture condition 1 (8.29 pH units and $p\text{CO}_2$ of 200 ppmv) and ±96.9 ppmv for culture condition 2 (7.55 pH units and $p\text{CO}_2$ of 1400 ppmv). These uncertainties were already accounted for in the final $\Delta\delta^{13}C_{\text{small-large}}$ / $CO_2$ equations through Monte Carlo simulations.

Meinrath, G., Spitzer, P. (2000). Uncertainties in Determination of pH. Mikrochim Acta 135, 155–168. doi.org/10.1007/s006040070005.

Changes in the revised manuscript (Section Materials and Methods) :

The pH values were checked by electrometric measurement using a Mettler Toledo pH InLab® Routine pro-ISM electrode. Calibration was performed using a TRIS buffer solution T34 (Del Valls & Dickson, 1998).

Del Valls, T.A. and Dickson, A.G. (1998). The pH of buffers based on 2-amino-2-hydroxymethyl-1,3-propanediol (''tris'') in synthetic sea water. Deep-Sea Res. 1 (45), 1541–1554.

The residual errors (also accounting for pH/$CO_2$ uncertainties) are evaluated through Monte Carlo analysis code with 1,000,000 iterations and an uncertainty of 0.17‰ for the differential vital effect between small and large coccolithes $\Delta\delta^{13}C_{\text{small-large}}$ (1σ).

Cell size is probably not the only factor influencing the degree of vital effects, as otherwise there should be similar ones found for H. carterii as more or less the same size as C. braarudii (see also next point).

We agree with the second comment from referee 2. The importance of collectively considering growth rate, PIC, POC, and the PIC:POC ratio in the expression of vital effects is already acknowledged in the manuscript (see lines 55 and 84). There is indeed no direct correlation between cell size and the magnitude of vital effects. Consequently, *H. carterii* and *C. braarudii* do not exhibit the same vital effect, as these species have distinct growth rates and PIC/POC ratios.

Line 55 reads : *However, when calcite is biomineralised intracellularly, biological parameters such as growth rate, cell size, and the PIC/POC ratio - which refers to the distribution of carbon between particulate organic carbon (POC) and particulate inorganic carbon (PIC) produced by calcifying organisms - also influence this fractionation (Dudley et al., 1986; McClelland et al., 2017; Rickaby et al., 2010).*

Line 84 reads : *Modelling studies fed by culture data have identified and quantified the main forcing parameters behind the magnitude of carbon isotope vital effect in coccolith calcite: growth rate, cell size, the partitioning of $CO_2$ in particulate inorganic matter and particulate organic matter (PIC/POC ratio), among other ancillary parameters (McClelland et al., 2017).*

For both quotations (and elsewhere in the ms), we thus believe we have already acknowledged that not only cell size was at play for the expression of the isotopic vital effects, but also growth rates and PIC/POC ratios.

Metabolic responses (growth and carbon fixation rates) of coccolithophores to increasing CO2 and decreasing pH follow optimum curves. If such curve is actually observed or not depends on the CO2 range chosen. The sensitivity to low CO2 levels (left hand side of the curve) and high proton concentrations (right hand side of the curve) is species and even strain specific (e.g. Langer et al. 2009). Furthermore, the response is modulated by other environmental factors such as light and temperature (e.g. Sett et al. 2014, Gafar et al. 2018). Hence, the pCO2 range within which vital effects are being displayed or not will vary. It is therefore unlikely, that a single transfer function derived for a particular combination of abiotic culturing conditions will be sufficient to reconstruct paleo pCO2 in a variable paleo environment (temperature, light, ....). Furthermore, and on top of potential strain-related vital effects, it would need to be shown that the off-set between small and large species is constant for different temperature and light conditions. All these caveats should properly be discussed and conclusions should be more cautious.

We agree with this view point. In this study, only the carbonate chemistry of the culture medium was changed, keeping other parameters constant: light irradiance and temperature as pointed by the Reviewer. Previous studies have also revealed a strain-specific biogeochemical response, especially for the morphotypes of *Emiliania huxleyi*. We will make these caveats more explicit in the revised manuscript.

In Section *4.3.2 Palaeoclimate application of carbon isotope culture data*, we already stated the following sentence: *One promising research avenue would be the study of the co-variation of $CO_2$/pH levels crossed with temperature changes in new culture campaigns. This approach could have the potential to revel the synergistic effect of discrete various controls on cell growth rates and refine the biogeochemical understanding of the vital effects*.

We will include a preliminary sentence at the beginning of the discussion (which will be recalled in the conclusion) so that the reader can appreciate the complexity of reproducing the natural environment (both geographically temporally) in experimental studies. We will include the suggested references.

Thus, we will add: "In this study, we chose to perturb only the carbonate chemistry of the culture medium in which the cells grew. It is important to remember that other environmental factors, such as light irradiance and temperature, also influence cellular growth and the magnitude of vital effects (Langer et a., 2009; Sett et al., 2014; Hermoso et al., 2016; Gafar et al., 2018)".

&

"The extent to which our biogeochemical calibration can be applied to wild and fossil coccoliths depends on the (paleo)environmental context. In these experiments, only the biogeochemical responses of monoclonal strains have been examined, while variations in light irradiance and temperature may interact with changes in the carbonate chemistry of water masses. In the future, a broader range of strains and additional physico-chemical parameters must be studied, integrating these factors to develop a unified response through modelling".

The array of pH conditions applied in the cultures is appreciable (7.55 – 8.29 pH units) and encompasses, to our best knowledge, most of the variation of this parameters in present day oceans. The investigated pH range also covers the documented variations of oceanic pH over the last 66 MYrs: 7.5 in the Eocene, 8.2 in the Pleistocene (Rae et al., 2021). The full spectrum of the pH/$CO_2$ forcing on physiological and isotopic responses by coccolithophore cells is therefore documented here.

Rae J.W.B., Zhang Y.G., Liu X., Foster G.L., Stoll H.M. and Whiteford R.D.M. (2021). Atmospheric CO2 over the Past 66 Million Years from Marine Archives. Annu Rev Earth Planet Sci, 49, 609-641. https://doi.org/10.1146/annurev-earth-082420-063026

Comparing the two transfer functions to derive paleo pCO2 from inorganic d13C measurements of large and small coccoliths shown in Figure 9 highlights the complications outlined above. For the same pCO2, they show differences of 1-2 per mille, meaning that, based on isotopic measurements, the reconstructed pCO2 would be off by a few hundred ppmv. Such high uncertainty should be acknowledged.

As mentioned to the reviewer 1 during the first round of peer-review, it is noteworthy that the two calibrations match pretty well, at least to first order (see Figure A1 for data over the entire $p\mathrm{CO_2}$/pH interval). The offset between the two studies can indeed originate from the above-mentioned points and/or the experimental setup. Replicated measurements within a much more restricted spread of values in our study gives confidence to the calibration (once again, in the conditions in which the cultures were performed). Our study has the advantage of having more measurements at low $p\mathrm{CO_2}$ compared to the work of Rickaby et al., 2010. This was the rationale for our culture work here. However, we acknowledge that there is a difference up to 1.5 ‰ between our calibration and that of Rickaby et al., 2010 at low $p\mathrm{CO_2}$ and high pH. The differences between the two calibrations highlight the idea exposed by referee 2 (in their third remark), i.e. we need more data from multi-stressor experiments / strains to refine the transfer equation. Implementing very dilute semi-continuous culture batch is essential for generating such biogeochemical dataset and avoiding a reservoir effect.

We propose adding the following paragraph at the end of Section 4.3.1:

A comparison of our calibration with the data published by Rickaby et al. (2010) in Figure 9 reveals a discrepancy of up to 1.5‰ under the lowest $p\mathrm{CO_2}$ / highest pH conditions. Explaining this difference is challenging, particularly because the studies were conducted under different conditions, involved different strains, and included only a few data points in this low $\mathrm{CO_2}$ range, as documented by Rickaby et al. (2010). This observation may call into question the feasibility of achieving a species- and environment-integrated response in coccolithophores (see further discussion in Section 4.3.2).

---

## Author Response (AR3)

Dear Pr. Middelburg,

Thank you for accepting of our article for publication, and for the suggested changes. Your three comments have been taken into account and the manuscript has been amended.

Please note that we have also slightly modified the wording of a sentence in the acknowledgements as follows:

We acknowledge with thanks the financial support from the French Agence Nationale de la Recherche (ANR) – Project CARCLIM under reference ANR-17-CE01-0004-01 to MH and from the Graduate school IFSEA under reference ANR-21-EXES-0011 (France 2030 programme). This work is part of the Graduate school IFSEA that benefits from grant ANR-21-EXES-0011 operated by the French National Research Agency under France 2030 program.

Sincerely,

Goulwen Le Guevel

Initial message :

Dear Dr. Le Guevel:

Thank you for submitting your revised version to Biogeosciences. I have read it with pleasure and I am happy to inform you that your paper is now accepted for publication. However, while reading this last version I identified three technical correction.

-p.7, line 169: please provide centrifuge intensity in G-forces (next to rpm).

-p.13, line 296: the word originally does not make sense for a reader. Please delete

-p.23, line 509: please replace revel with reveal.

With best regards, Jack Middelburg, Associate editor